# Estimating fine age structure and time trends in human contact patterns from coarse contact data: The Bayesian rate consistency model

Shozen Dan [1,©,¤,‡,*], Yu Chen [1,©,¤,‡], Yining Chen[1], Melodie Monod[1], Veronika K. Jaeger [2], Samir Bhatt[3,4], André Karch[2], Oliver Ratmann [1,©,¤,*], on behalf of the Machine Learning & Global Health network

**1** Department of Mathematics, Imperial College London, London, England, United Kingdom, **2** Institute of Epidemiology and Social Medicine, University of Münster, Münster, Germany, **3** School of Public Health, Imperial College London, London, England, United Kingdom, **4** Department of Public Health, University of Copenhagen, Copenhagen, Denmark

© These authors contributed equally to this work.
¤ Current Address: Imperial College London, Exhibition Road, London, United Kingdom
‡ These authors share first authorship on this work.
* shozen.dan21@imperial.ac.uk (SD); oliver.ratmann@imperial.ac.uk (OR)

**Data Availability Statement:** The data and code required to reproduce all the results presented in this scientific paper are publicly available from the

## Abstract

Since the emergence of severe acute respiratory syndrome coronavirus 2 (SARS-CoV-2), large-scale social contact surveys are now longitudinally measuring the fundamental changes in human interactions in the face of the pandemic and non-pharmaceutical interventions. Here, we present a model-based Bayesian approach that can reconstruct contact patterns at 1-year resolution even when the age of the contacts is reported coarsely by 5 or 10-year age bands. This innovation is rooted in population-level consistency constraints in how contacts between groups must add up, which prompts us to call the approach presented here the Bayesian rate consistency model. The model can also quantify time trends and adjust for reporting fatigue emerging in longitudinal surveys through the use of computationally efficient Hilbert Space Gaussian process priors. We illustrate estimation accuracy on simulated data as well as social contact data from Europe and Africa for which the exact age of contacts is reported, and then apply the model to social contact data with coarse information on the age of contacts that were collected in Germany during the COVID-19 pandemic from April to June 2020 across five longitudinal survey waves. We estimate the fine age structure in social contacts during the early stages of the pandemic and demonstrate that social contact intensities rebounded in an age-structured, non-homogeneous manner. The Bayesian rate consistency model provides a model-based, non-parametric, computationally tractable approach for estimating the fine structure and longitudinal trends in social contacts and is applicable to contemporary survey data with coarsely reported age of contacts as long as the exact age of survey participants is reported.

GitHub repository at the following URL: https://github.com/MLGlobalHealth/bayes-rate-consistency.

**Funding:** This work was funded by the Imperial President's PhD Scholarship program and the EPSRC Centre for Doctoral Training in Modern Statistics and Statistical Machine Learning at Imperial and Oxford (EP/S023151/1) to YC and MM. The Bill & Melinda Gates Foundation (OPP1175094) to MM and OR. The Medical Research Council (MR/V038109/1) to OR. The MRC Centre for Global Infectious Disease Analysis (MR/R015600/1), jointly funded by the UK Medical Research Council (MRC) and the UK Foreign, Commonwealth & Development Office (FCDO), under the MRC/FCDO Concordat agreement, and also part of the EDCTP2 programme supported by the European Union and the Novo Nordisk Foundation via The Novo Nordisk Young Investigator Award (NNF20OC0059309) and the Danish National Research Foundation via a chair position and the Eric and Wendy Schmidt Fund For Strategic Innovation via the Schmidt Polymath Award (G-22-63345) and the National Institute for Health Research (NIHR) via the Health Protection Research Unit in Modelling and Health Economics to SB. COVIMOD is funded by intramural funds of the Institute of Epidemiology and Social Medicine, University of Munster, and of the Institute of Medical Epidemiology, Biometry and Informatics, Martin Luther University Halle-Wittenberg, as well as by funds provided by the Robert Koch Institute, Berlin, the Helmholtz-Gemeinschaft Deutscher Forschungszentren e.V. via the HZEpiAdHoc "The Helmholtz Epidemiologic Response against the COVID-19 Pandemic" project, the Saxonian COVID-19 Research Consortium SaxoCOV (co-financed with tax funds on the basis of the budget passed by the Saxon state parliament), the Deutsche Forschungsgemeinschaft (DFG, German Research Foundation, via the project SpaceImpact project number 458526380) and the Federal Ministry of Education and Research (BMBF) via the projects Respinow (project number 031L0298F) and OptimAgent (project number 031L0299J) and as part of the Network University Medicine (NUM) via the egePan Unimed project (funding code: 01KX2021). The funders had no role in study design, data collection and analysis, the decision to publish, or the preparation of the manuscript.

**Competing interests:** The authors have declared that no competing interests exist.

## Author summary

The transmission of respiratory infectious diseases occurs during close social contacts. Hence, measuring the intensity and patterns in social contacts leads to a better understanding of disease spread and provides essential data to estimate central quantities such as the reproduction number in real-time. Unlike pre-pandemic surveys, which largely recorded contacts' age in one-year age intervals, most COVID-era studies only recorded the age of contacts in broad age categories to facilitate reporting. Some studies allowed participants to report an estimate for the total number of contacts for which they could not remember age and gender information. Many studies were partially longitudinal, which introduced the issue of reporting fatigue. Thus, directly applying existing statistical methods for estimating social contact matrices may result in losing age detail and confounded estimates. To this end, we develop a model-based approach which estimates fine-age contact patterns from coarse-age data by exploiting particular constraints that must hold mathematically in closed populations. The model can also adjust for the confounding effects of aggregate contact reporting and reporting fatigue and estimate the time trends in social contact dynamics. We hope this statistical model is a useful addition to the global pandemic preparedness toolkit to reconstruct the fine structure of social contact patterns and measure real-time effective reproduction numbers with greater precision.

This is a *PLOS Computational Biology* Methods paper.

## Introduction

The transmission of human respiratory diseases such as influenza, tuberculosis, and COVID-19 is directly driven by the rate of close social contact between individuals. Social contact studies such as the pivotal POLYMOD study [1] have been widely acknowledged as an effective method of obtaining social contact estimates to assess infection risk and to parameterise mathematical infectious disease models [2–4]. Consequently, they provide critical epidemiological insights which inform the implementation and evaluation of non-pharmaceutical interventions [5] as well as public health policies such as vaccination schedules [6].

Since the outbreak of COVID-19, longitudinal social contact surveys have been conducted in Europe and around the world, providing indispensable information on the evolving patterns of human mixing behaviour during the pandemic [7, 8]. In Germany, the COVIMOD study collected social contact data for nearly two years, and initial analyses [9] focused on data from April to June 2020 during the first partial lockdown in Germany to quantify the scale of social contact reductions relative to pre-pandemic contact patterns. It is important to estimate high-resolution age- and gender-specific human contact patterns and their trends over time, but this has remained an unsolved challenge due to changes in survey methodology from pre-pandemic studies [1, 9] and limitations in available inference methods [2, 10, 11].

First, COVIMOD and other COVID-era social contact studies record the age of contacts by large age bands of 5 to 10 years, reflecting that often it is difficult for study participants to know the exact age of their contacts. In contrast, the pre-pandemic POLYMOD surveys primarily collected data on the exact age of contacts. Subsequently developed methods, including thin-plate regressions [2] and Gaussian Markov Random Field model approaches [10] relied

on such data to estimate high-resolution contact patterns, and these methods are no longer directly applicable. Most studies have thus resorted to the convenient and speedy bootstrap approach implemented in the `socialmixr` library [11], although it can only provide contact estimates in the same coarse age bands that the data are reported in, which mask the full structure of the actual contact patterns [12].

Second, most COVID-era studies adopted retrospective web-based survey protocols and conducted longitudinal repeat surveys [7, 13–15]. The survey waves are typically inter-dependent because a number of participants were surveyed in multiple waves, and additional participants were recruited to replenish the cohort size. While this approach provides valuable longitudinal data, it also introduces the issue of reporting fatigue, where participants tend to report fewer contacts in subsequent participation due to becoming tired of filling out the survey. It follows that directly applying existing methods, which do not incorporate adjustments to counter the confounding reporting fatigue effects, is bound to lead to incorrect estimates. Additionally, participants sometimes found it difficult to recall specific age and gender information for all of their contacts. Instead, they were allowed to report an estimate for the total number of contacts on that occasion [9], which again may result in under-ascertainment of contact intensities if these data are not accounted for in inference approaches of contact patterns.

This work presents a non-parametric Bayesian model to infer age- and gender-specific contact patterns and time trends at high 1-year resolution from longitudinal survey data. The primary innovation of the model is the ability to infer contact patterns by 1-year age bands even when the age contacts are reported in broad age bands. We call this model-based approach the Bayesian rate consistency model for reasons that will be clear soon. In addition, we use recently developed Hilbert Space Gaussian Process approximations [16] to gain substantial advances in computational efficiency, enabling us to make full Bayesian inferences over time and uncover the time trends in social contact structure. We demonstrate that it is crucial to model contact patterns over time to account for reporting fatigue effects in inter-dependent longitudinal survey waves. The primary purpose behind developing the Bayesian rate consistency model is its application to contemporary COVID-19-era survey data, which we present for data spanning the first five survey waves of the COVIMOD study in Germany. We present high-resolution estimates of age- and gender-specific social contacts for each survey wave and describe their time evolution. We also place the inferred contact dynamics into a pre-pandemic context and quantify the differences in contact intensity change by the age of contacts.

## Methods

### Ethics statement

COVIMOD was approved by the ethics committee of the Medical Board Westfalen-Lippe and the University of Münster, reference number 2020–473-fs. Written consent was obtained from all participants. The POLYMOD data collection was approved by national institutional review boards [1]. As only anonymised COVIMOD and POLYMOD data were used in this study, an institutional review was not required for reanalysis.

### The COVIMOD study

The COVIMOD study was launched in April 2020 and continued until December 2021, constituting 33 survey waves. Participants were recruited through email invitations to existing panel members of the online market research platform IPSOS i-say [17]. To ensure the sample's broad representativeness of the German population, quota sampling was conducted based on age, gender, and region. Participants were invited to participate in multiple waves to track changes in social behaviour and attitudes toward COVID-19. When the participant size

did not meet the sampling quota due to study withdrawals, new participants were recruited into the study. This approach enabled the COVIMOD study to obtain longitudinal samples, but it also introduced the issue of response fatigue, where the number of detailed contacts reported decreased compared to previous participation, irrespective of the survey wave. To procure information on children, a subgroup of adult participants living with children under the age of 18 were selected to be proxies. This procedure meant that middle-aged adults were under-sampled as they completed the survey on behalf of their children.

The COVIMOD questionnaire was based on the CoMix study and includes questions on demographics, the presence of a household member belonging to a high-risk group, attitudes towards COVID-19 as well as related government measures, and current preventative behaviors [9, 18]. Participants were also asked to provide information about their social contacts between 5 a.m. the preceding day to 5 a.m. the day of answering the survey. Following the pre-pandemic POLYMOD study, a contact is defined as either a skin-to-skin contact such as a kiss or a handshake (physical contact) or an exchange of words in the presence of another person (non-physical contacts) [1]. Participants were asked to report the age group, gender, relation, the contact setting (e.g. home, school, workplace, place of entertainment, etc.), and whether the contact was a household member. For survey waves 1 and 2, participants were asked to provide each contact's information separately. However, some participants reported contacts to groups of individuals (e.g., customers, clients) for which a specific number of contacts was assumed (Additional file 2 of [9]). From wave 3 onward, in addition to being asked to report contacts individually, participants were allowed to report a total number of contacts that were either physical or non-physical contacts in the sense described above and which they did not report individually, which we refer to as aggregate contacts. Additionally, some participants could not recall or preferred not to answer the age or gender information of some of the individual contacts that they reported. We treat these three types of entries with missing age or gender equally and refer to them as *missing & aggregate contact reports*. A copy of the COVI-MOD questionnaire may be found in Additional file 1 of [9].

This current work concerns the first five survey waves of the COVIMOD study. In Fig 1A and 1B, we show the sampling periods with the number of daily COVID-19 cases, cumulative COVID-19-related deaths, and the OxCGRT Stringency Index which is a measure that captures the degree of containment and closure policies in place for COVID-19 on a scale of 0% to 100% [19]. The following COVID-19 policy timeline is obtained from the ACAPS COVID-19 Government Measures dataset [20]. The first COVIMOD survey was administered from April 30th to May 6th in the year 2020, towards the end of the first partial lockdown and the first wave of cases. Before the beginning of the first survey (April 20th), small stores, auto dealers, and bookstores were allowed to reopen under strict hygiene regulations. During the final few days of the survey period (May 4th to 6th), phase-out measures were announced by the government, including the step-wise uptake of schools, the reopening of hairdressers under strict hygiene regulations, lifting of the ban on public gatherings of 30 people indoors and 50 outdoors, resumption of religious services, and reopening of public services such as museums, botanical gardens, zoos, and playgrounds. The second wave of the COVIMOD survey was administered from May 14th to May 21st. During this period, additional phase-out measures were announced, including the resumption of all cross-country transport and the reopening of hotels and restaurants. International travel to neighbouring countries was also slightly relaxed during this period. The third, fourth, and fifth waves of COVIMOD surveys were taken from May 28th to July 4th, June 11th-22nd, and June 26th to July 1st, respectively. There was no notable introduction or reduction of social contact restriction measures during this time, but international travel restrictions were relaxed primarily for Schengen and EU countries. COVID-19 cases and deaths remained stable during this period (Fig 1A and 1B).

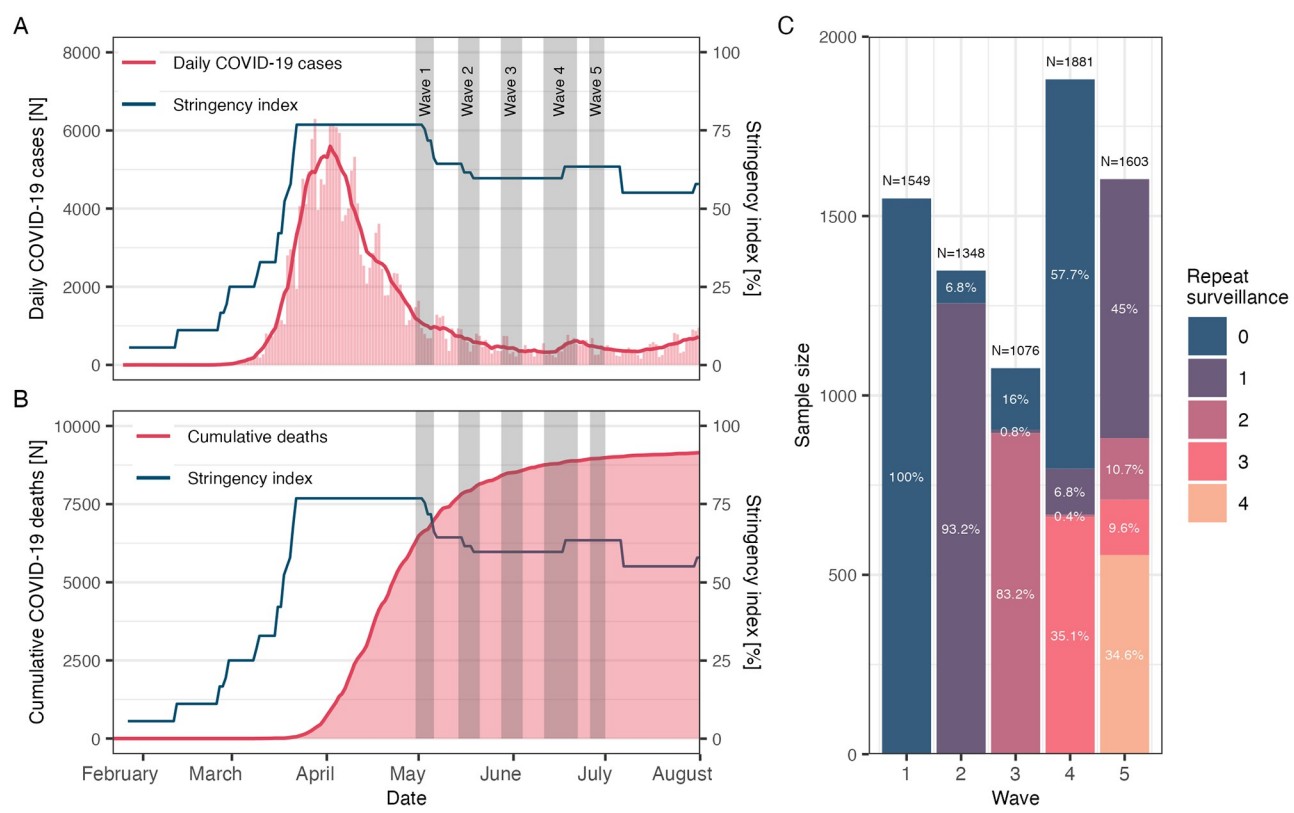

**Fig 1. Timeline and participants of the longitudinal COVIMOD study.** A. Daily COVID-19 case counts in Germany (red bars), the OxCGRT Stringency Index (blue line), and COVIMOD survey administration periods (grey ribbons). B. Cumulative COVID-19-related deaths in Germany (red line), the OxCGRT Stringency Index (blue line), and COVIMOD survey administration periods (grey ribbons). C. Sample sizes and the proportion of people repeatedly sampled in the COVIMOD survey for which zero repeats indicate first-time participants.

After excluding participants who prefer not to provide age or gender information and 25 participants above the age of 84, there were 1549, 1345, 1076, 1881, and 1603 participants for waves 1 to 5. We observed 3244, 4852, 6344, 13471, and 8353 total contacts for each wave. In Fig 1C, we show the proportion of participants who consented to the survey multiple times. Most participants in waves 2 and 3 had participated in wave 1, with only 6.8% and 16% of participants being new to the survey. The proportion dropped sharply in wave 4, where only 35.1% of initial participants remained. Hence the majority (57.7%) of wave 4 participants were first-time participants. On the contrary, no new participants were enrolled for wave 5, and individuals who participated for the second, third, and fifth time took up approximately 45%, 10.7%, 9.6%, and 34.6% of the sample.

## Data processing

In this study, we excluded reports from 20 participants (0.3% of the total) who did not report their own age or gender, as these data are essential for our modelling framework. Further, and following ethical guidelines, the participant age information for children was reported in discrete age bands, i.e., 0–4, 5–9, 10–14, 15–18 years. To obtain fine-age information for participants under 18, we imputed their age by drawing from a discrete uniform distribution with bounds set as the minimum and maximum age of the participant's age category. Finally, regarding the three types of missing & aggregate contacts reported, i.e. contacts reported to

groups of individuals, additional contacts reported for who participants did not provide individual information, and individual contacts reported for who participants did not provide age and gender information, we truncated these at 60 (90th percentile) to remove the effects of extreme outliers. We show the distribution of observed contacts in S1 and S2 Figs.

## Estimating contact patterns

Throughout, we focus on estimating contact patterns and dynamics between men and women (superscripts $g, h \in \{M, F\}$) and high resolution one year age groups (subscripts $a, b \in \mathcal{B} = \{0, 1, 2, \cdots 84\}$). We note that the cutoff at age 84 is a specific choice on account of the age aggregation scheme and the age-specific sample size of COVIMOD. One may change this cutoff for different surveys in an appropriate fashion. To introduce basic notations, let us momentarily consider male-to-female contacts at some fixed time $t$, suppress the time index, and assume that the exact age of reported contacts is known. We start by considering the total number of contacts $Y_{ab}^{MF}$ that are reported by men of age $a$ who participate in the survey to all women of age $b$ in the population. The number of male survey participants of age $a$ is $N_a^M$, and the number of women of age $b$ in the population is $P_b^F$. When $N_a^M > 0$ for age group $a$, the contact counts $Y_{ab}^{MF}$ are defined for all $b \in \mathcal{B}$, and are either zero or positive. When there are no participants for some age group $a$, there is no corresponding contact data, and we denote the participant age groups in the survey by $\mathcal{A}^M = \{a \in \mathcal{B}: N_a^M > 0\}$. Finally, we denote the number of age groups in $\mathcal{A}^M$ by $A^M$ and the number of age groups in $\mathcal{B}$ by $B$. We also consider analogous notations for all other gender combinations using the superscripts $g$ and $h$.

From the observed data, we seek to estimate the *contact intensity* $m_{ab}^{MF}$, the average number of contacts from one male participant of age $a$ to all women aged $b$ in the population within a 24-hour time window. We follow [1, 2, 8, 10, 21–23] and model the count data with an over-dispersion adjusting Negative Binomial observation model, in shape-scale form,

$$Y_{ab}^{gh} \sim \text{NegBinomial}\left(\alpha_{ab}^{gh}, \frac{v}{1+v}\right) \tag{1a}$$

$$\mu_{ab}^{gh} = \alpha_{ab}^{gh} v \tag{1b}$$

$$\log \mu_{ab}^{gh} = \log m_{ab}^{gh} + \log N_a^g \tag{1c}$$

where $g, h \in \{M, F\}$, $a \in \mathcal{A}^g$, $b \in \mathcal{B}$. Here, $\mu_{ab}^{gh}$ are the expected contacts, which are expressed in terms of the target quantity of interest, $m_{ab}^{gh}$, and the known $N_a^g$. The overdispersion parameter $v$ is strictly greater than zero ($v > 0$), such that the variance of the contact counts $\text{Var}[Y_{ab}^{gh}] = \mu_{ab}^{gh}(1 + v)$ is greater than the mean.

The *contact rate* is defined as the probability of contact between one male aged $a$ and one female aged $b$ within a 24-hour time window, i.e.

$$\gamma_{ab}^{MF} = m_{ab}^{MF}/P_b^F. \tag{2}$$

Crucially, in a closed population, the number of total contacts between the same population strata must be self-consistent and add up to the same, i.e.

$$P_a^M P_b^F \gamma_{ab}^{MF} = P_b^F P_a^M \gamma_{ba}^{FM}, \tag{3}$$

for all $a, b \in \mathcal{B}$. From this, we find contact rates are symmetric because $\gamma_{ab}^{MF} = \gamma_{ba}^{FM}$ for all $a$, $b$, and similarly $\gamma_{ab}^{MM} = \gamma_{ba}^{MM}$ and $\gamma_{ab}^{FF} = \gamma_{ba}^{FF}$ for all $a < b$. Property (3) implies that data on age group $a$ informs contact rates in both age dimensions, which we will exploit heavily below. If

the survey captures participants for all possible age groups, i. e. $\mathcal{A}^g = \mathcal{B}$, the estimation problem reduces to $B \times B + B \times (B + 1)/2 \times 2 = B(2B + 1)$ free contact rate parameters rather than $4B^2$ free parameters, an almost 50% reduction. To take advantage of these self-consistency constraints, we follow [10] and expand Eq (1c) to

$$\log m_{ab}^{gh} = \beta_0 + f^{gh}(a, b) + \log(P_b^h), \quad g = M, \ h = F, \ a, b \in \mathcal{B}, \tag{4a}$$

$$\log m_{ab}^{hg} = \beta_0 + f^{gh}(b, a) + \log(P_b^g), \quad g = M, \ h = F, \ a, b \in \mathcal{B}, \tag{4b}$$

$$\log m_{ab}^{gg} = \beta_0 + f^{gg}(a, b) + \log(P_b^g), \quad g \in \{M, F\}, \ a \le b, \tag{4c}$$

$$\log m_{ab}^{gg} = \beta_0 + f^{gg}(b, a) + \log(P_b^g), \quad g \in \{M, F\}, \ a > b, \tag{4d}$$

where $\beta_0 \in \mathbb{R}$ is a real-valued baseline parameter, and $f^{MF}, f^{MM}, f^{FF}$ are three real-valued, random functions of two-dimensional continuous inputs on the compact domain $[0, 84] \times [0, 84]$. Specifically, we model the $f^{gh}$ through computationally efficient Gaussian process approximations as described below, and for ease of notation, write $f^{gh}(a, b)$. The random functions act as age-age-specific offsets to the baseline parameter and thus capture the age structure in human contact intensities. Using random functions, we can estimate arbitrary age-specific contact patterns. This is important because human contact patterns have changed substantially since the COVID-19 pandemic with school closures and other non-pharmaceutical interventions.

## Recovering fine age structure from coarse data

For COVIMOD and similar contact surveys, participants were asked to report their contacts in the coarse age groups

$$c \in \mathcal{C} = \{0 - 4, \ 5 - 9, \ 10 - 14, \ 15 - 19, \ 20 - 24, \ 25 - 34, \ 35 - 44,$$
$$45 - 54, \ 55 - 64, \ 65 - 69, \ 70 - 74, \ 75 - 79, \ 80 - 84\}. \tag{5}$$

Importantly, the exact age of the participant is known, and we can leverage this information through the symmetry property in Eq (3) to estimate contact intensities at a much finer resolution. Because of this fundamental property, we call our resulting model the "Bayesian rate consistency model". Using the shape-scale parameterisation in Eq (1), it follows that

$$Y_{ac}^{gh} = \sum_{b \in c} Y_{ab}^{gh} \sim \text{NegBinomial}\left( \sum_{b \in c} \alpha_{ab}^{gh}, \frac{v}{1 + v} \right) \tag{6a}$$

$$\mu_{ab}^{gh} = \alpha_{ab}^{gh} v \tag{6b}$$

$$\log \mu_{ab}^{gh} = \log m_{ab}^{gh} + \log N_a^g, \tag{6c}$$

where $g, h \in \{M, F\}$, $a \in \mathcal{A}^g$, $b \in \mathcal{B}$, and $c \in \mathcal{C}$. We will demonstrate below that the high-resolution contact intensities $m_{ab}^{gh}$ are identifiable from coarse contact data.

## Estimating dynamics in contact patterns

It is in principle, straightforward to extend model (1–6) to capture time trends in contact patterns, but a particular challenge arises in the context of repeat surveillance. Across COVIMOD

survey waves, many participants agreed to report data in multiple rounds, and analyses indicate that participants tend to repeat fewer contacts in subsequent surveys, a phenomenon called reporting fatigue. The time trends in the primary data are thus confounded by longitudinal reporting behaviour. To control for reporting fatigue, we denote by $Y_{trac}^{gh}$ the number of contacts to individuals of age group $c$ and gender $h$ that are reported in survey wave $t$ by all participants of age $a$ and gender $g$ who have participated $r$ time(s). All other notation extends analogously.

In the simplest case, we introduce age-homogeneous reporting fatigue effects $\rho_r \in \mathbb{R}$ at repeat response times $r = 0, 1, 2, \ldots$, and jointly model the longitudinal data with

$$Y_{trac}^{gh} \quad \sim \text{NegBinomial}\left(\sum_{b \in c} \alpha_{trab}^{gh}, \frac{v}{1+v}\right) \tag{7a}$$

$$\mu_{trab}^{gh} \quad = \alpha_{trab}^{gh} v \tag{7b}$$

$$\log \mu_{trab}^{gh} \quad = \log m_{tab}^{gh} + \rho_r + \log\left(N_{tra}^g\right), \tag{7c}$$

where $t = 1, 2, \ldots$ indicates the survey waves, $r = 0, 1, 2, \ldots$ repeat surveillance, and $\rho_0 = 0$ for ease of notation. Here, reporting fatigue is captured by negative $\rho_r$ (that decreases with $r$), which in turn will adjust the contact intensities $\log m_{tac}^{gh}$ in follow-up survey rounds to higher estimates than the primary data suggest. New participants entered the COVIMOD survey in each survey wave, so we have data to provide independent information on the contact dynamics and reporting fatigue, with the model borrowing strength across all the data available.

A second challenge is that the number of missing & aggregate contact reports also fluctuated over time. As denominator in our sample, we now consider all participants of age $a$ and gender $g$ regardless of whether they reported individual contacts or missing & aggregate contacts and denote their total in wave $t$ with repeat participation $r$ by $N_{tra}^g$ as in (7c). We further denote the number of missing & aggregate contact reports by participants of age $a$ and gender $g$ in survey wave $t$ by $T_{ta}^g$. For instance, if a participant in wave $t$ of age $a$ and gender $g$ had 2 contacts with missing age or information and reported 18 aggregate contacts, then we added 20 to $T_{ta}^g$. In addition, consider the number of total contacts with detailed age information by participants of age $a$ and gender $g$ in survey wave $t$ by $Y_{ta}^g = \sum_{r,c,h} Y_{trac}^{gh}$. Thus, we can calculate the proportion of contacts that are reported with detailed age information,

$$S_{ta}^g = Y_{ta}^g / \left(Y_{ta}^g + T_{ta}^g\right). \tag{8}$$

Assuming that the age of contacts is missing at random within each participant group of gender $g$, age $a$ and wave $t$, we then use (8) as an additional offset term in the linear predictor,

$$\log \mu_{trab}^{gh} = \log m_{tab}^{gh} + \rho_r + \log\left(N_{tra}^g\right) + \log\left(S_{ta}^g\right). \tag{9}$$

In practice, if increasingly many participants only provide aggregated contact reports, we have that $S_{ta}^g < 1$, and in turn, this will adjust the contact intensities $\log m_{tab}^{gh}$ in later survey waves to higher estimates than the data with full age-specific details suggest.

## Non-parametric modelling of contact dynamics

We regularise our inferences in high-dimensional parameter space by associating the random functions $f_t^{MF}, f_t^{MM}, f_t^{FF}$ in Eq (4) with computationally efficient, zero-mean, two-dimensional Hilbert Space Gaussian Process priors [24, 25]. In what follows, we will drop the time and

gender sub- and superscripts to ease notation and present our modelling of a generic random function $f$ that represents age structure in contact patterns. Zero-mean two-dimensional Gaussian Processes (GPs) are powerful prior models for random functions. For any finite collection of two-dimensional inputs, the function values are multivariate normal with mean zero. We always have $AB$ count observations on the grid $x_1 = (a_1, b_1), \ldots, x_{AB} = (a_A, b_B)$ defined by $A$ male and female participant age groups in $\mathcal{A}$ and all possible $B$ population age groups in $\mathcal{B}$. The multivariate normal has then a covariance matrix $\boldsymbol{K} \in \mathbb{R}^{AB \times AB}$ whose $i, j^{\text{th}}$ entries are specified by a covariance kernel function $k(x_i, x_j)$. Here, we decompose the 2D kernel function for computational efficiency and model each component through squared exponential or Matérn class kernels. Specifically, using the squared exponential kernels as an example, we have

$$k((a, b), (a', b')) = k^1(a, a')k^2(b, b') \tag{10a}$$

$$k^1(a, a') = \alpha_a^2 \exp\left(-\frac{(a - a')^2}{2l_a^2}\right) \tag{10b}$$

$$k^2(b, b') = \alpha_b^2 \exp\left(-\frac{(b - b')^2}{2l_b^2}\right), \tag{10c}$$

where the scaling parameters $\alpha_a$, $\alpha_b$ control the magnitude of the random function in the corresponding dimension, and the lengthscale parameters $l_a$, $l_b$ control the bandwidth. The product in Eq (10a) is also known as Kronecker decomposition because the covariance matrix $\boldsymbol{K}$ equals the Kronecker product of the covariance matrices of the kernels with one-dimensional inputs, $\boldsymbol{K} = \boldsymbol{K}^2 \otimes \boldsymbol{K}^1$, where the $i, j$th entry in $\boldsymbol{K}^1 \in \mathbb{R}^{A \times A}$ is given by $k^1(a_i, a_j)$ and the $i, j$th entry in $\boldsymbol{K}^2 \in \mathbb{R}^{B \times B}$ is given by $k^2(b_i, b_j)$. For computing purposes, we exploit that $\boldsymbol{K}^1$, $\boldsymbol{K}^2$ are positive semi-definite and decompose the covariance matrices as $\boldsymbol{K}^1 = \boldsymbol{L}^1\boldsymbol{L}^{1\top}$, $\boldsymbol{K}^2 = \boldsymbol{L}^2\boldsymbol{L}^{2\top}$, where the superscript $\top$ denotes transposition. Using the mixed product property of Kronecker operations, we obtain

$$\boldsymbol{K} = (\boldsymbol{L}^2 \otimes \boldsymbol{L}^1)(\boldsymbol{L}^2 \otimes \boldsymbol{L}^1)^\top. \tag{11}$$

This shows that the zero-mean two-dimensional GP prior attached to the random function $f$ on the $AB$ inputs $\boldsymbol{x} = (x_1, \ldots, x_{AB})$ can be obtained by linear transformation of $AB$ i. i. d. standard Gaussian random variables $\boldsymbol{z} \sim \mathcal{N}(0, 1)$,

$$\boldsymbol{f}(\boldsymbol{x}) = (\boldsymbol{L}^2 \otimes \boldsymbol{L}^1)\boldsymbol{z} = \text{vec}\left(\left(\boldsymbol{L}^2\left(\boldsymbol{L}^1 \, \text{reshape}(\boldsymbol{z}, A, B)\right)^\top\right)^\top\right). \tag{12}$$

In Eq (12), the left-hand side denotes the $AB$-dimensional column vector of the random function evaluated at the inputs, and the right-hand side shows how the Kronecker product is calculated by a series of basic arithmetic operations. The reshape operation transforms the $AB$ dimensional column vector $\boldsymbol{z}$ column-wise into a $A \times B$ dimensional matrix, and the vec operation flattens $A \times B$ dimensional matrices column-wise into an $AB$ dimensional column vector.

Eq (12) also shows that the computational cost of two-dimensional GPs is entirely determined by calculating, first, $\boldsymbol{L}^1$, $\boldsymbol{L}^2$ for each new set of GPs hyperparameters, and then, second, performing the arithmetic operations associated with $(\boldsymbol{L}^2 \otimes \boldsymbol{L}^1)\boldsymbol{z}$. We use Hilbert Space Gaussian Process (HSGP) approximations [16] to each of the kernels $k^1$ and $k^2$ in Eq (10) to reduce the computational cost associated with the first step. For brevity, we refer readers to the

excellent introductions to HSGPs in [24, 25], and here merely note that the stationary isotropic kernels can be expressed as an infinite sum that involves the spectral density $S$, eigenfunctions $\phi_i$ and eigenvalues $\lambda_i$, $i = 1, \ldots, \infty$, associated with a certain Laplacian eigenvalue problem on a compact domain $\Omega$ that is strictly larger than $\mathcal{B}$. For convenience, the input domain $\mathcal{B}$ is shifted with the midpoint at zero, and then $\Omega$ is written as $[-L, L]$ for some $L > 0$. To ease notation, we continue to write the shifted inputs in what follows as $a_i$, and $b_i$. The HSGP approximation $\tilde{k}^1$ to $k^1$ on the domain $[-L^1, L^1]$ is then obtained by truncating the infinite sum to the first $M^1$ terms,

$$k^1(a, a') \approx \tilde{k}^1(a, a') \quad = \sum_{j=1}^{M^1} S^1(\sqrt{\lambda_j^1})\phi_j^1(a)\phi_j^1(a'), \tag{13a}$$

$$S^1(\omega) \quad = \alpha_a^2(2\pi l_a)\exp(-l_a^2\omega^2/2), \tag{13b}$$

$$\sqrt{\lambda_j^1} \quad = (j\pi)/(2L^1), \tag{13c}$$

$$\phi_j^1(x) \quad = \sqrt{1/L^1}\sin(\sqrt{\lambda_j^1}(x + L^1)). \tag{13d}$$

Crucially, the GP hyperparameters $\alpha_a$, $l_a$ enter only in Eq (13b), and the eigenvalues and eigenfunctions are the same regardless of the GP hyperparameters and depend only on the domain boundary value $L^1$ together with the observed inputs $a$, $a'$. This speeds up Bayesian computations significantly because the eigenvalues in Eq (13c) and eigenfunctions in Eq (13d) can be precomputed once and for all. For Matérn and other kernels, the corresponding spectral densities may be found in [26]. Rewriting Eq (13a) in matrix notation, we see that $\boldsymbol{L}^1$ is approximated by

$$\boldsymbol{L}^1 \approx \tilde{\boldsymbol{L}}^1 = \Phi^1\sqrt{\Delta^1}, \tag{14}$$

where the $A \times M^1$ matrix $\Phi^1$ has the $i, j$ entries $\phi_j^1(a_i)$, and the $M^1 \times M^1$ matrix $\Delta^1$ is diagonal with $j, j$ entries $S^1(\sqrt{\lambda_j^1})$. Again, $\Phi^1$ does not depend on the GP hyperparameters and can be precomputed. The arithmetic operations in Eq (14) can harness computationally efficient diagonal-post-multiply functions in many linear algebra libraries. The HSGP approximation to the $k^2$ kernel is analogous. The tuning parameters of the HSGP approximations are the integers $M^1$, $M^2$ and the boundary values $L^1$, $L^2$, and we determine these using established diagnostics [25]. The zero-mean Kronecker-decomposed HSGP prior associated with our random functions $\boldsymbol{f}$ on the input grid $\boldsymbol{x}$ is then

$$\boldsymbol{f}(\boldsymbol{x}) = (\tilde{\boldsymbol{L}}^2 \otimes \tilde{\boldsymbol{L}}^1)\tilde{z} = \mathrm{vec}\left( \left( \tilde{\boldsymbol{L}}^2 \left( \tilde{\boldsymbol{L}}^1 \, \mathrm{reshape}(\tilde{z}, M^1, M^2) \right)^\top \right)^\top \right), \tag{15}$$

where $\tilde{z}$ is a $M^1 M^2$ dimensional column vector of i. i. d. standard normal random variables, and the non-negative hyperparameters are $\theta = (\alpha_a, l_a, \alpha_b, l_b)$.

## Difference-in-age parameterisation

Human contact patterns tend to concentrate among individuals of similar age and individuals with similar age gaps (parent-child, grandparent-child and grandparent-parent). To capture this diagonal structure in the simple Kronecker decomposed priors in Eq (10) for our 2D

random functions $\boldsymbol{f}$, we follow [22] and define $\boldsymbol{f}$ on an age by difference-in-age space rather than an age by age space. This amounts to rotating the age by age space by 45 degrees so that the peer-peer, parent-child, grandparent-child, and grandparent-parent contacts correspond to horizontal lines in the re-parameterised space and match the structure of our Kronecker decomposed priors (10) (see also Fig 1 of Vandendijck et al. [22]). Specifically, we consider age differences $d \in \mathcal{D} = \{-84, -83, \ldots, 83, 84\}$, and re-parameterise the points $(a, b) \in \mathcal{A} \times \mathcal{B}$ to $(a, d) = d(a, b) = (a, b - a) \in \mathcal{A} \times \mathcal{D}$. We are only interested in the random functions evaluated on the original points, which we write as $\boldsymbol{f}(d(a, b))$ for all $(a, b) \in \mathcal{A} \times \mathcal{B}$. The number of age differences $D = 169$ in $\mathcal{D}$ is larger than the number of one-year age groups $B = 85$ and so the difference-in-age parameterisation entails higher computational cost in the calculations that underpin Eq (15). We will show in the results section that this parameterisation is crucial to obtaining accurate estimates of typical fine-age diagonal and off-diagonal human contact patterns from coarsely reported age data, and so recommend its use despite the higher computational cost.

## Full Bayesian model and numerical inference

To complete our model for inferring contact dynamics from longitudinal survey data, we specified commonly used priors on all model parameters. The full model is specified for survey waves $t = 1, \ldots, 5$, reporting repeats $r = 0, \ldots, 4$, gender $g, h \in \{M, F\}$, participant age groups $a \in \mathcal{A}^{trg}$ and population age groups $b \in \mathcal{B}$ by

$$Y_{trac}^{gh} \sim \text{NegBinomial}\left(\sum_{b \in c} \alpha_{trab}^{gh}, \frac{v}{1+v}\right) \tag{16a}$$

$$\mu_{trab}^{gh} = \alpha_{trab}^{gh} v \tag{16b}$$

$$\log \mu_{trab}^{gh} = \log m_{tab}^{gh} + \rho_r + \log\left(N_{tra}^g\right) + \log\left(S_{ta}^g\right) \tag{16c}$$

$$\log m_{tab}^{gh} = \beta_0 + \tau_t + \boldsymbol{f}_t^{gh}(d(a, b)) + \log\left(P_b^h\right), \quad g = M, \ h = F, \ a, b \in \mathcal{B} \tag{16d}$$

$$\log m_{tab}^{hg} = \beta_0 + \tau_t + \boldsymbol{f}_t^{gh}(d(b, a)) + \log\left(P_b^g\right), \quad g = M, \ h = F, \ a, b \in \mathcal{B} \tag{16e}$$

$$\log m_{tab}^{gg} = \beta_0 + \tau_t + \boldsymbol{f}_t^{gg}(d(a, b)) + \log\left(P_b^g\right), \quad g \in \{M, F\}, \ a \leq b, \tag{16f}$$

$$\log m_{tab}^{gg} = \beta_0 + \tau_t + \boldsymbol{f}_t^{gg}(d(b, a)) + \log\left(P_b^g\right), \quad g \in \{M, F\}, \ a > b, \tag{16g}$$

and

$$\beta_0 \sim \mathcal{N}(0, 10) \tag{17a}$$

$$\rho_r \sim \mathcal{N}(0, 1) \tag{17b}$$

$$\tau_t \sim \mathcal{N}(0, 1) \tag{17c}$$

$$v \sim \text{Exponential}(1) \tag{17d}$$

$$\boldsymbol{f}_t^{gh}(d(\boldsymbol{x}))|\alpha_{ti}, l_{ti} \sim \text{HSGP}(\boldsymbol{0}, \tilde{\boldsymbol{L}}_t^{gh,2} \otimes \tilde{\boldsymbol{L}}_t^{gh,1}), \quad gh \in \{MF, MM, FF\}, i = 1, 2 \tag{17e}$$

$$\alpha_{ti} \sim \text{Cauchy}^+(0, 1), \quad i = 1, 2 \tag{17f}$$

$$l_{ti} \sim \text{InvGamma}(5, 5), \quad i = 1, 2. \tag{17g}$$

Monte Carlo draws from the joint posterior distribution of all parameters was obtained with the probabilistic computing language Stan [27] via the cmdstanr interface version 0.5.2. Eight chains were run in parallel for 500 warmup iterations and 1000 iterations thereafter. Initial sampling was facilitated by adding the nugget $10^{-13}$ to $\alpha_{trab}^{gh}$. We typically observed a small number of divergences in the NUTS algorithm, but these accounted for less than 0.005% of samples and were considered to be of no concern. The smallest effective sample size was 1892, and the $\hat{R}$ convergence diagnostics were below 1.01, indicating that the Markov chains converged and mixed well [28, 29]. Trace plots for the parameters with the smallest effective sample size and maximum $\hat{R}$ are shown in S8 Fig. The R and Stan codes for our models, along with other analysis codes are available at https://github.com/MLGlobalHealth/bayes-rate-consistency.

## Simulated social contact data

To validate the Bayesian rate consistency model, we created synthetic datasets that mimic social contact patterns before the COVID-19 pandemic (pre-COVID-19) and during the pandemic (in-COVID-19). To reduce experiment run time, we limited participants and contact ages from 6 to 49 years and assumed that contact intensity patterns do not vary by gender.

Contact intensities were set to be highest among individuals of similar age, mimicking age-assortative contact behaviour. To simulate parent-child contact dynamics, we defined individuals between 6–18 as children and individuals between 30–39 as parents and set the contact intensities between parent-child groups to higher values. Similarly, individuals between 19–29 were defined as children of individuals from 40 to 49, and contact intensities between these groups were also set to a higher value. The resulting patterns are shown in the top left panel of Figs 2 and 3. For full details, we refer readers to S1 Text. From the stylised contact intensity scenarios, we next randomly generated age- and gender-specific contact counts for five different participant size configurations, $N = 250, 500, 1000, 2000, 5000$, by sampling from a Poisson distribution such that $Y_{ab}^{gh} \sim \text{Poisson}(\lambda_{ab}^{gh})$ where $\lambda_{ab}^{gh} = \tilde{m}_{ab}^{gh} N_a^g$. We set $N_a^g$ such that the age-gender counts of the participants were representative of the 2011 German census population [30]. To mimic the age reporting scheme in the COVIMOD surveys, we aggregated the simulated contact counts by $Y_{ac}^{gh} = \sum_{b \in c} Y_{ab}^{gh}$ where $c \in \{$6-9, 10-14, 15-19, 20-24, 25-34, 35-44, 45-49$\}$ as illustrated in the top right panels of Figs 2 and 3. In this fashion, we generated 10 replicate datasets for each experiment configuration (pre-COVID-19/in-COVID-19 and sample size) to obtain representative accuracy and runtime estimates.

## Results

### Contact patterns by 1-year age band can be estimated

Fig 2 illustrates the simulated contact patterns in the pre-COVID-19 scenario by 1-year age bands, along with the corresponding data by coarse 5 and 10-year age bands for a sample size of 2,000 participants, and the fits with the Bayesian rate consistency model that aim to recover the fine age structure by 1-year age bands. Fig 3 shows our results for the in-COVID-19 scenario. The age-age parameterisation performed poorly for both simulation scenarios, especially in regions of the contact matrix where the degree of age aggregation is large, i.e., 10-year age intervals as opposed to 5-year age intervals. For such large reporting intervals of age groups, the contact intensity patterns that we could estimate with the age-age parameterisation showed idiosyncratic bimodal patterns along the main diagonal of the contact intensity matrices (bottom left panels in Figs 2 and 3). In comparison, the difference-in-age parameterisation captured age-assortative contact patterns and the sub-diagonal parent-children contact patterns with much better accuracy. The estimated contact intensity patterns for other gender combinations and simulation scenarios were qualitatively very similar and reported in S4 and S5 Figs.

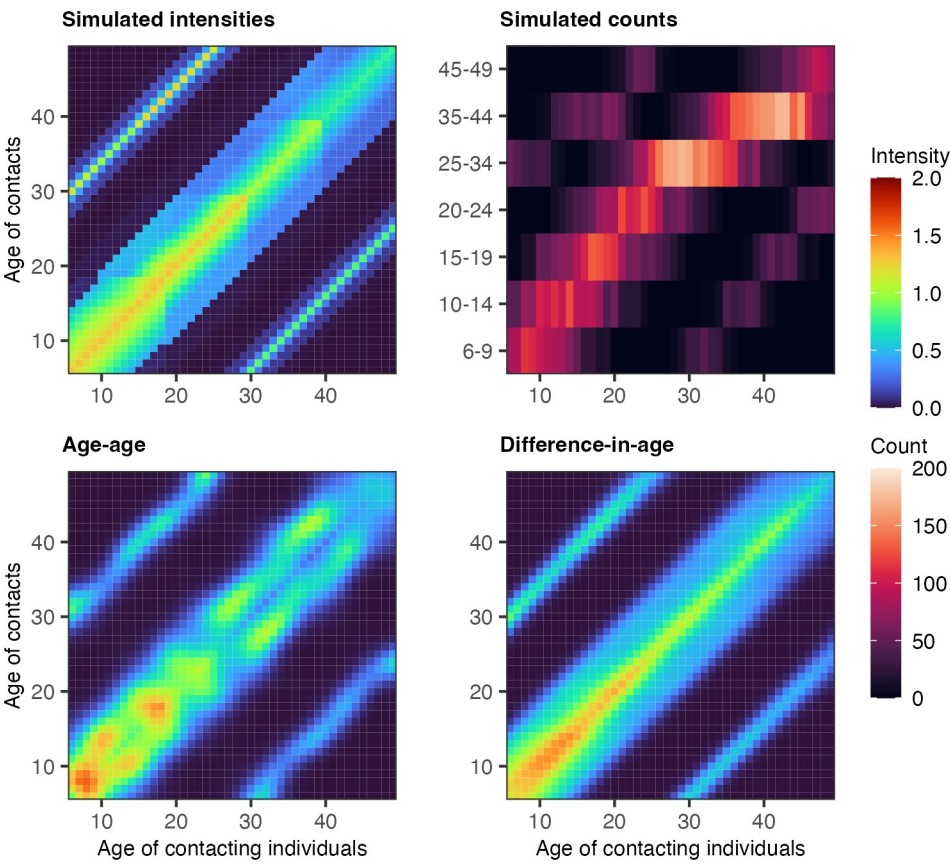

**Fig 2. Pre-COVID19 scenario simulation experiments.** (Top left) Simulated social contact intensities for Male-Male contacts. (Top right) Simulated social contact counts for Male-Male contacts with a COVIMOD-like age aggregation scheme. (Bottom left) Posterior median estimates of social contact intensities from the age-age parameterised HSGP model. (Bottom right) Posterior median estimates of social contact intensities from the difference-in-age parameterised HSGP model.

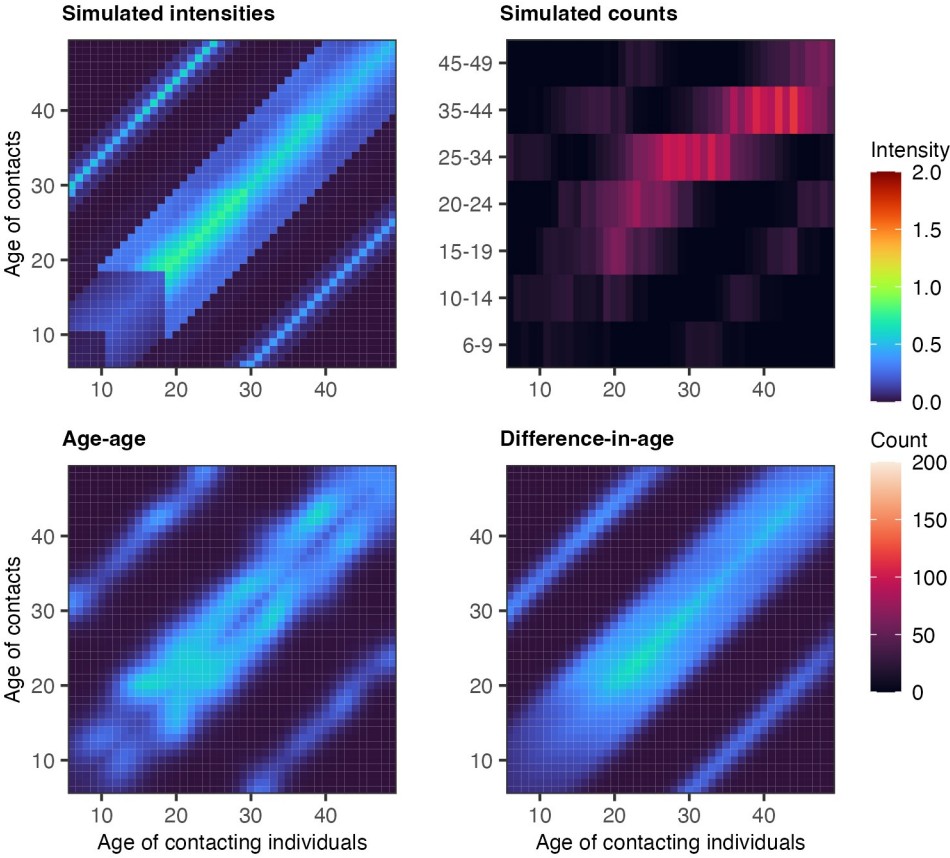

**Fig 3. In-COVID19 scenario simulation experiments.** (Top left) Simulated social contact intensities for Male-Male contacts. (Top right) Simulated social contact counts for Male-Male contacts with a COVIMOD-like age aggregation scheme. (Bottom left) Posterior median estimates of social contact intensities from the age-age parameterised HSGP model. (Bottom right) Posterior median estimates of social contact intensities from the difference-in-age parameterised HSGP model.

In several previous social contact surveys, participants were asked to report the exact age of their contacts. This allowed us to further test the Bayesian rate consistency model by artificially withholding the exact age of each contact in real-world data. Specifically, we tested how closely the reconstructed fine-age contact intensities inferred on the masked data with the age of contact masked into broad age bands match those inferred from the actual data with the exact age of contacts known. Fig 4 shows the outcomes of this test on data from 1,292 participants who reported 31,670 contacts in Germany as part of the POLYMOD study [1]. The contact intensities learned from the data with the exact age of contact and those learned from the data with age of contact masked into broad age bands were qualitatively very similar, and the mean absolute difference across each age-age gender-gender group was 0.145 in the context of a mean contact intensity of 0.120 in each age-age gender-gender group. We repeated this test on data from 1,122 participants from the Manicaland HIV/STD Prevention Study [31], a general population cohort study carried out since 1998 in Manicaland, the easternmost province of Zimbabwe. In 2013, participants were asked to report on their social contacts, which were either physical or non-physical, in two-way conversations with three or more words in the physical presence of another person, and the exact age was reported for 24,480 contacts. We chose these data for a second test because Manicaland, Zimbabwe's population structure and social

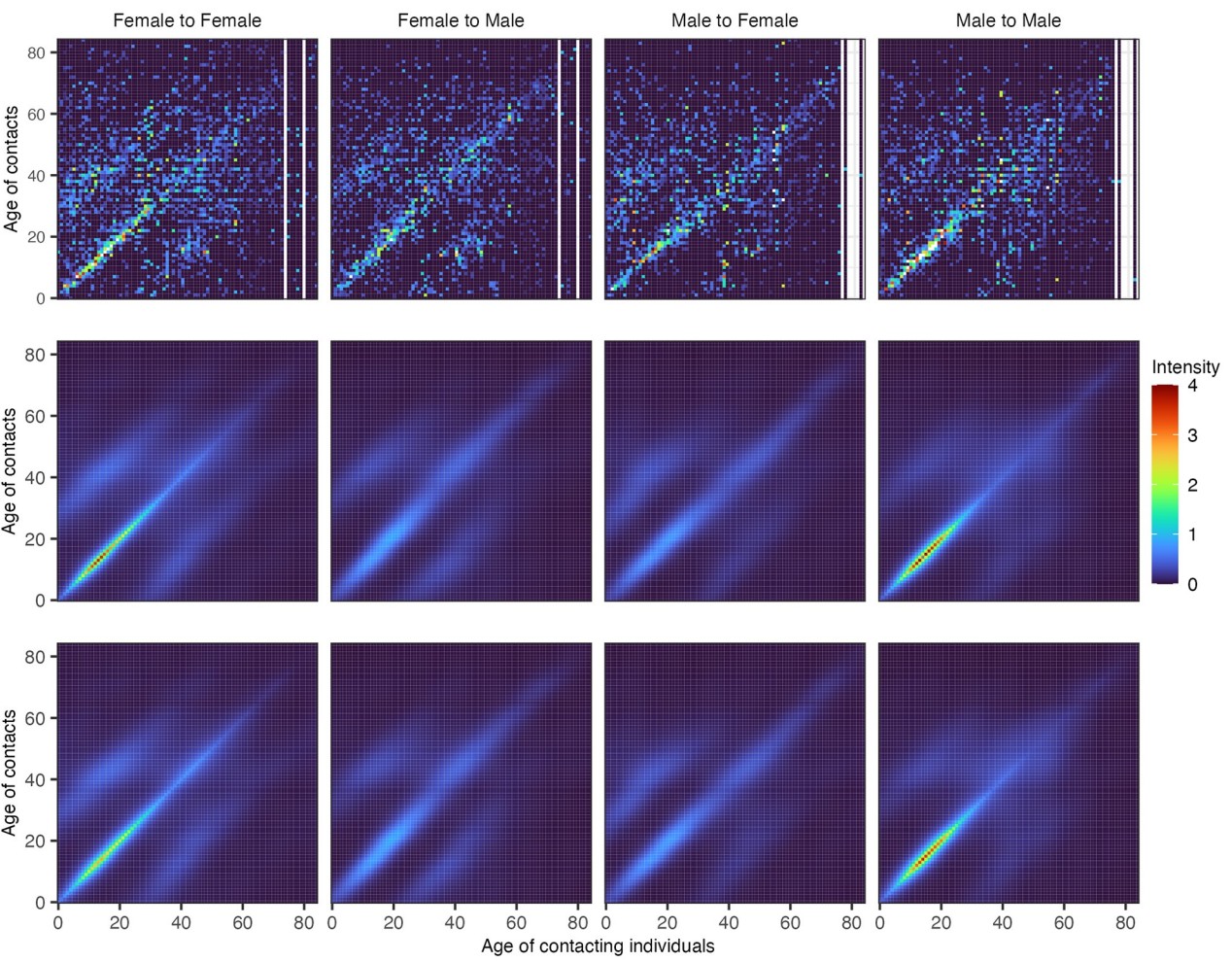

**Fig 4. Empirical and estimated contact intensity patterns for POLYMOD using the Bayesian rate consistency model for age-granular contact data and age-stratified contact data.** (Top row) Crude empirical contact intensity patterns. (Middle row) Posterior median contact intensity estimates from the Bayesian rate consistency model applied to fine-age contact data meaning the age of contacts was not aggregated into large age bands. (Bottom row) Posterior median contact intensity estimates from the Bayesian rate consistency model applied to coarse-age contact data meaning the age of contacts was aggregated into larger age bands similarly to the COVIMOD study.

contact patterns differ markedly from those seen in Europe [1, 31]. Fig 5 shows our test outcomes on the Manicaland data. The Bayesian rate consistency model generated again very similar estimates on the data with the age of contact masked into broad age bands as compared to the actual data with the exact age of contact known, and the mean absolute difference across each age-age gender-gender group was 0.179 in the context of a mean contact intensity of 0.125 in each age-age gender-gender group.

## Gaussian process approximations enable fast Bayesian inference

In Table 1 and S6 and S7 Figs, we compare the performance of the Bayesian rate consistency model with various parameterisations for different sample sizes and scenarios in terms of estimation accuracy and computing runtimes. Specifically, to assess how well HSGP models can approximate full-rank 2DGP models, we ran simulations for both scenarios and parameterisations with a sample size fixed at 2000. We compared the model fits to the simulation truth in

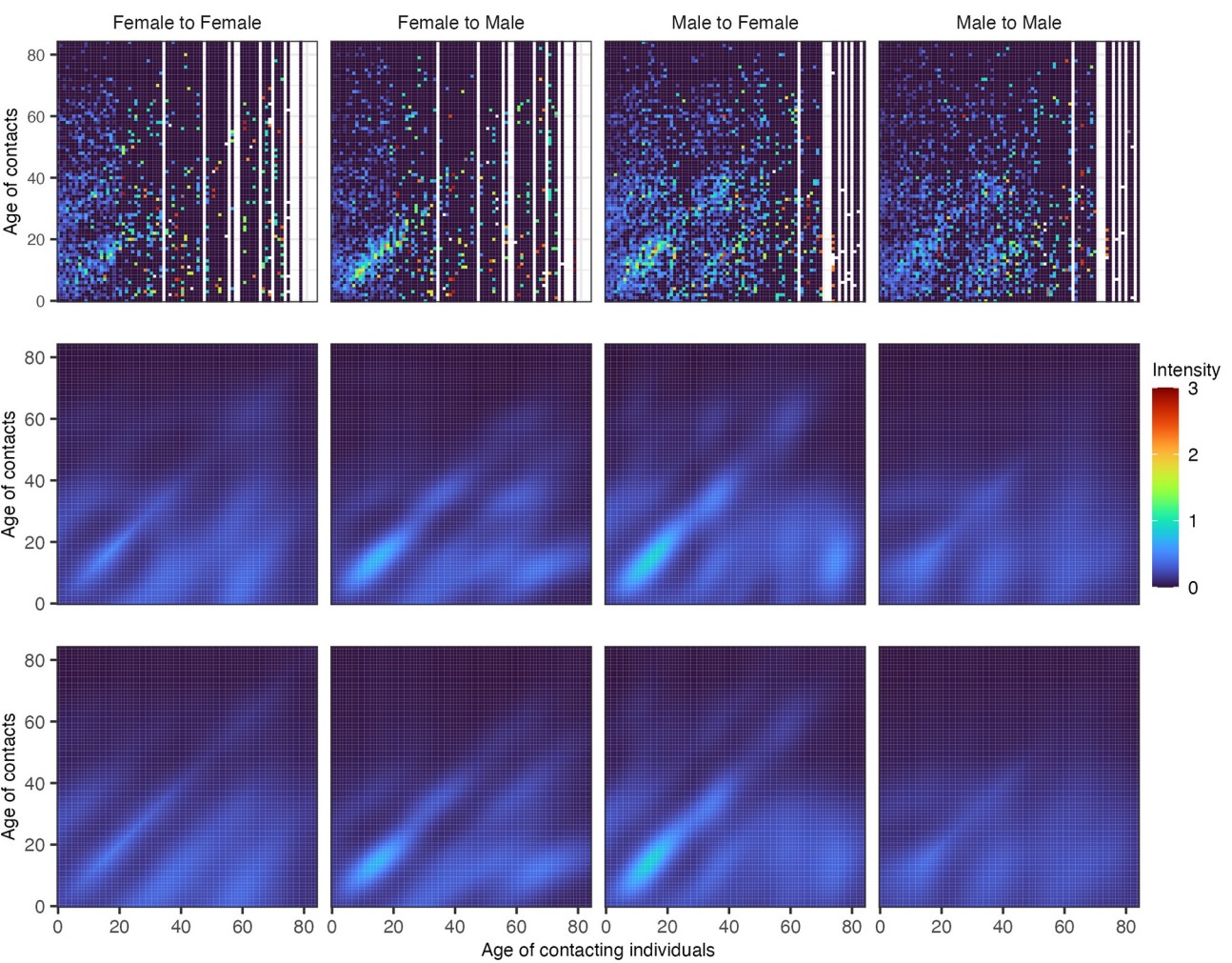

**Fig 5. Empirical and estimated contact intensity patterns for Zimbabwe using the Bayesian rate consistency model for age-granular contact data and age-stratified contact data.** (Top row) Crude empirical contact intensity patterns. (Middle row) Posterior median contact intensity estimates from the Bayesian rate consistency model applied to fine-age contact data (i.e., the age of contacts is not aggregated into large age bands). (Bottom row) Posterior median contact intensity estimates from the Bayesian rate consistency model applied to coarse-age contact data (i.e., the age of contacts is aggregated into large age bands in a similar fashion to COVIMOD).

terms of the mean absolute error (MAE) based on the age-age-specific inferred contact intensities, expected log posterior density (ELPD), the percentage that the predicted values are inside the 95% prediction intervals according to posterior predictive check (PPC), and the median running time.

The difference-in-age parameterisation achieved significantly better accuracy than the age-age parameterisation but also required more time to fit due to the introduction of additional $A^2 - A$ nuisance parameters. The computational toll of the difference-in-age parameterisation was most strongly reflected in the median runtimes under the full-rank 2DGPs, which could take more than half a day to fit. Using HSGPs, we reduced median runtimes by more than 15-fold while maintaining an accuracy close to that of full-rank 2DGP models. The bottom section of Table 1 compares the accuracy of the difference-in-age HSGP models across survey sample sizes. In general, smaller sample sizes led to less accurate estimates.

**Table 1. Comparison of performance on simulated data for different scenarios, models, sample sizes, and parameterisations.**

| Scenario | N | Model | Parameterisation | MAE[a] | ELPD[b] | PPC[c] | Runtime[d] |
|---|---|---|---|---|---|---|---|
| pre[e] | 2000 | 2DGP | age-age | $8.27 \times 10^{-2}$ | -3762.1 | 99.9% | 0.6 hours |
| pre | 2000 | HSGP | age-age | $8.39 \times 10^{-2}$ | -3737.2 | 99.7% | 0.2 hours |
| pre | 2000 | 2DGP | difference-in-age | $5.19 \times 10^{-2}$ | -3607.1 | 98.4% | 15.4 hours |
| pre | 2000 | HSGP | difference-in-age | $5.12 \times 10^{-2}$ | -3622.3 | 98.7% | 1.0 hours |
| in[f] | 2000 | 2DGP | age-age | $4.47 \times 10^{-2}$ | -3117.9 | 99.9% | 1.0 hours |
| in | 2000 | HSGP | age-age | $4.52 \times 10^{-2}$ | -3099.9 | 99.7% | 0.2 hours |
| in | 2000 | 2DGP | difference-in-age | $3.24 \times 10^{-2}$ | -3015.1 | 98.6% | 9.8 hours |
| in | 2000 | HSGP | difference-in-age | $3.17 \times 10^{-2}$ | -3017.0 | 98.8% | 1.0 hours |
| pre | 5000 | HSGP | difference-in-age | $5.46 \times 10^{-2}$ | -4738.4 | 98.2% | 1.0 hours |
| pre | 2000 | HSGP | difference-in-age | $5.12 \times 10^{-2}$ | -3622.3 | 98.7% | 1.0 hours |
| pre | 1000 | HSGP | difference-in-age | $5.32 \times 10^{-2}$ | -3029.6 | 98.7% | 1.1 hours |
| pre | 500 | HSGP | difference-in-age | $5.52 \times 10^{-2}$ | -2497.8 | 98.8% | 1.0 hours |
| pre | 250 | HSGP | difference-in-age | $5.71 \times 10^{-2}$ | -1915.5 | 99.0% | 0.9 hours |
| in | 5000 | HSGP | difference-in-age | $3.11 \times 10^{-2}$ | -3958.2 | 98.4% | 1.0 hours |
| in | 2000 | HSGP | difference-in-age | $3.17 \times 10^{-2}$ | -3017.0 | 98.8% | 1.0 hours |
| in | 1000 | HSGP | difference-in-age | $3.34 \times 10^{-2}$ | -2504.6 | 99.0% | 1.0 hours |
| in | 500 | HSGP | difference-in-age | $3.45 \times 10^{-2}$ | -1875.4 | 99.1% | 1.2 hours |
| in | 250 | HSGP | difference-in-age | $4.92 \times 10^{-2}$ | -1428.1 | 99.4% | 1.0 hours |

[a]Mean absolute error,

[b]Expected log posterior density,

[c]Posterior predictive check,

[d]Median runtime,

[e]pre-COVID19 scenario,

[f]in-COVID19 scenario.

## Modelling marked structures in age-specific contact patterns

Social contacts are strongly structured by age, reflecting common behaviour and social norms around family size, reproductive age, schooling, and other factors [22]. In turn, smooth process kernels such as the squared exponential (Eq (10b)) may not be well suited to describe marked changes in contact intensities. On our simulated contact scenarios, we find indeed that Matérn $\frac{5}{2}$ and Matérn $\frac{3}{2}$ kernels performed better in comparison to squared exponential kernels in terms of accuracy (S1 Table). The difference in accuracy between Matérn $\frac{3}{2}$ and Matérn $\frac{5}{2}$ was small and qualitatively indistinguishable (S4 and S5 Figs), and in the following results, we considered the Matérn $\frac{5}{2}$ kernel.

## Model-based estimates of contact patterns in Germany by 1-year age groups

Fig 6 shows the crude empirical age- and gender-specific contact intensities for the first wave of the COVIMOD survey in early May 2020, calculated from the data without any statistical modelling via

$$\hat{m}_{ac}^{gh} = \frac{Y_{ac}^{gh}}{N_a^g} \frac{1}{S_a^g},$$

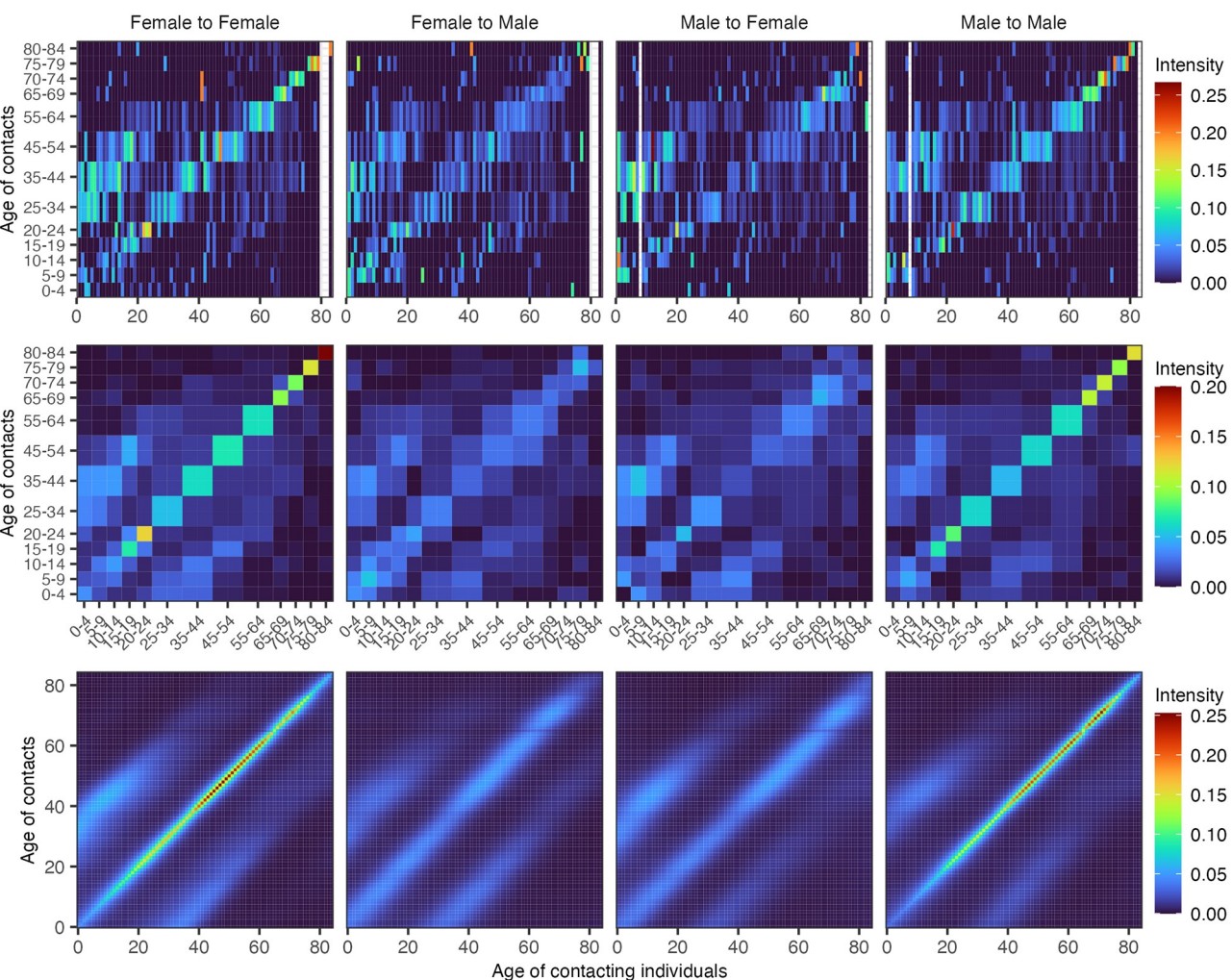

**Fig 6. Empirical and estimated contact intensity patterns for COVIMOD wave 1.** (Top row) Crude empirical social contact intensity patterns. There are some age groups with no participants, and they are represented by white vertical columns. (Middle row) Contact intensity patterns as estimated by the `socialmixr` R package [11]. (Bottom row) Posterior median contact intensity estimates from the Bayesian rate consistency model. For the crude estimates and `socialmixr` estimates, estimates are divided by the number of integer ages in the corresponding age group to make the estimates more comparable (*e.g.*, if the age of contacts is $20 - 24$ than the contact estimate is divided by 5 while if the age of contacts is $25 - 34$ the contact estimate is divided by 10). The exact runtime arguments for this comparison are given in script `figure-6.R` on our accompanying GitHub repository.

where $Y_{ac}^{gh}$ denote the total number of contacts from participants of gender $g$ and age group $a$ to individuals of gender $h$ and age category $c$, $N_a^g$ are age- and gender-specific sample size, and $S_a^g$ are age- and gender-specific proportion of reports with complete age and gender information. Fig 6 also shows the contact intensity estimates obtained via bootstrapping with the `socialmixr` package [11], and the estimates from the Bayesian rate consistency model. We found that the crude estimates are sparse and fluctuate greatly, even between neighbouring age groups. Next, the `socialmixr` estimates aggregate the observations by the large age categories in which the contacts were reported and do not borrow available information through the exact age of participants to obtain higher resolution estimates. The `socialmixr` estimates are adjusted for symmetry in contact rates but are not adjusted for reporting fatigue or missing & aggregate contact reports (see respectively Eqs (7) and (9)). We also observed that contacts

with missing age information are imputed by sampling the missing age only from all contacts of the participants of the same age group [11].

In comparison, the contact patterns estimated with the Bayesian rate consistency model align with those estimated with `socialmixr`, but provide finer age resolution. We achieve this higher resolution by logical constraints on who contacts whom in a closed population (recall Eq (1)), and not via imputation. These constraints imply that data on the exact age of survey participants provide information on the exact age of contacts even though they are reported in coarse age brackets. The patterns reveal strong age-assortativeness in mixing patterns that is indicated by the high intensities on the main diagonals of the contact intensity matrices shown in the bottom row of Fig 6. Lying approximately 30 years away from the main diagonal, two strips of high contact intensity fade with increasing age and correspond to intergenerational contacts between parents and children. While this age-dependent pattern persists over time (S9 and S12 Figs), we will also show below that the increases in social contact intensities in subsequent waves were far from uniform across age.

## Controlling for time-varying reporting effects

We can sum the contacts' age dimension of estimated contact intensities to obtain the average number of contacts from one person of age *a* per day. For brevity, we call these "marginal" contact intensities. In Fig 7, we show the estimated marginal contact intensities under the Bayesian rate consistency model, which simultaneously accounts for the time-varying reporting effects that emerge through reporting fatigue (repeat participation in the longitudinal COVIMOD survey) and missing & aggregate contact reports (participants unable to list contacts individually), and which are clearly present in the data as shown in Figs 1 and S1. We compare these (adjusted) marginal contact intensities in Fig 7 to those obtained without adjusting for missing & aggregate contact reports (but adjusting for reporting fatigue), those obtained without adjusting for reporting fatigue (but adjusting for missing & aggregate contact reports), and those obtained without any adjustments for missing & aggregate contact reports or reporting fatigue. Furthermore, we compared the marginal contact intensities of wave 4 given by the final longitudinal model, which adjusts for reporting fatigue with a cross-sectional model fitted on data from the same wave but with repeating participants excluded from the data (S13 Fig). We found that the reporting effect sizes can be estimated simultaneously with all other age, gender and time parameters, meaning that they are mathematically consistent with the symmetry constraints in contact rates in closed populations. The results clearly show that adjusting for missing & aggregate contact reports and reporting fatigue significantly increased the estimated marginal contact intensities in a non-trivial manner that depends on the contribution of repeat survey participants to each survey wave. It is implausible that the actual contact intensities did not increase as contact reduction measures were progressively eased between wave 1 to wave 5 of the COVIMOD survey, as is suggested by the estimates without adjustments. This led us to conclude that longitudinal contact patterns must be estimated from the available data with a model-based framework that can adjust for fatigue effects and missingness.

## Social contact intensities from May to July 2020 in Germany remained largely below pre-pandemic levels

With these adjustments in place, we next compared the contact intensities seen during the first 5 waves of the COVIMOD study to those observed in the pre-pandemic POLYMOD study conducted between 2006 and 2008 [1] (Figs 8 and S14). Social contact intensities in the first 5 waves remained substantially below pre-pandemic levels, which illustrates the stark impact

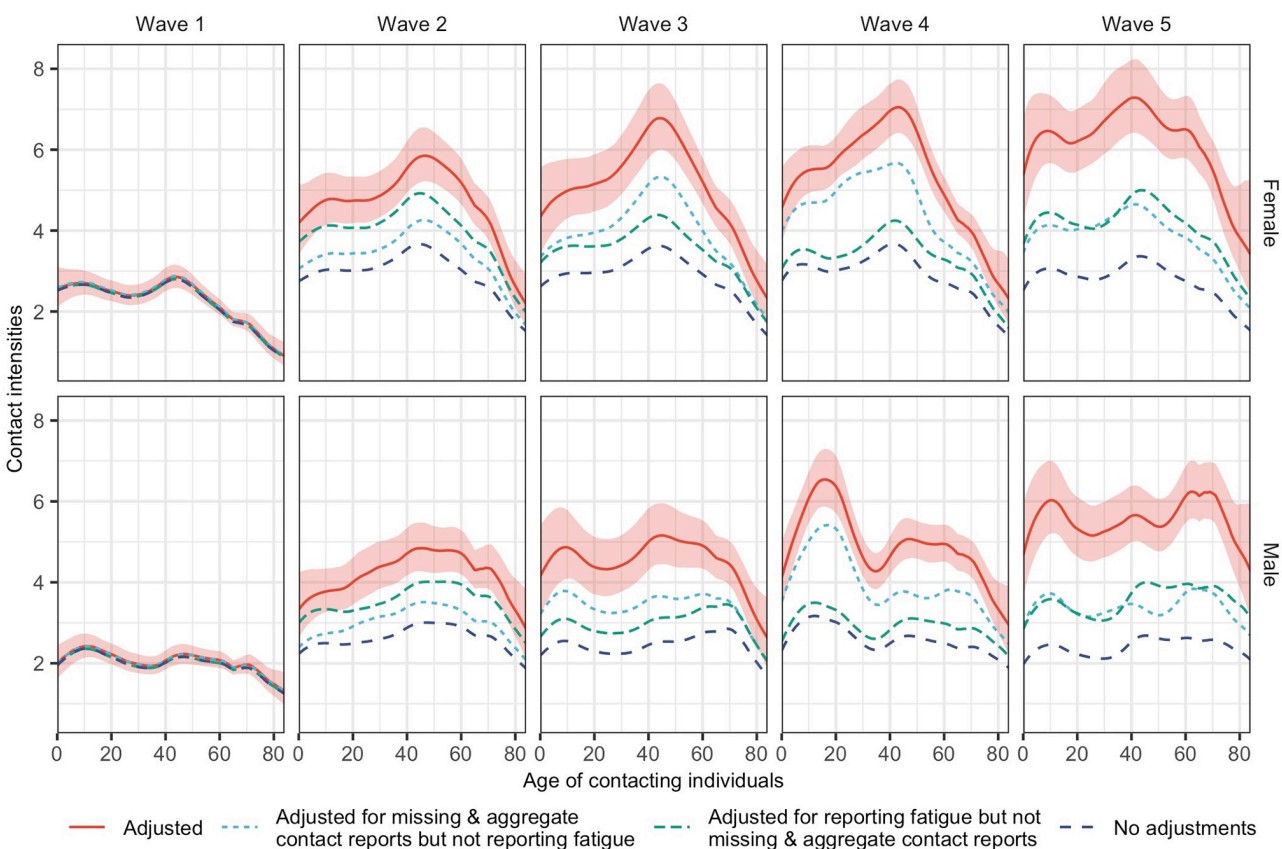

**Fig 7. Adjustments on marginal social contact intensities.** Posterior median estimates (line) and 95% credible intervals (ribbon) of the number of social contacts in the last 24 hours are shown by age and sex of participants and survey wave in red under the Bayesian rate consistency model that adjusts for both missing & aggregate contact reports and reporting fatigue. For comparison, posterior median estimates are shown for the model incorporating no adjustments (long dashed blue), adjustments for reporting fatigue but not missing & aggregated contact reports (green dashed lines), and adjustments for missing & aggregated contact reports but not reporting fatigue (turquoise small dashed lines).

that the COVID-19 pandemic and non-pharmaceutical interventions had on social interactions when protective vaccines were unavailable, and mortality following COVID-19 infection was high [32]. Interestingly, the POLYMOD data also suggest that women between ages 20 and 50 had more contacts than men of the same age range before the pandemic (Fig 8). Another point that emerges from the fine-age analysis is that by wave 5, the marginal contact intensities of individuals aged 70 were very similar to pre-pandemic levels. This could reflect that the relatively few social contacts of older individuals aged 70 and above are essential human interactions that are challenging to reduce.

## Non-homogeneous increases in contact intensities from May to July 2020 in Germany

Fig 7 shows that the contact intensities increased consecutively from wave 1 to wave 5, with the increases being most substantial from wave 1 to wave 2. These increases are primarily due to marked increases in the number of contacts that were reported in aggregate over the first five survey waves (S1–S3 Figs). Fig 9 illustrates the relative percentage increase in the marginal contact intensities relative to those in wave 1. Although there were marked differences in the contact patterns between men and women (Fig 7), the relative increases showed no significant

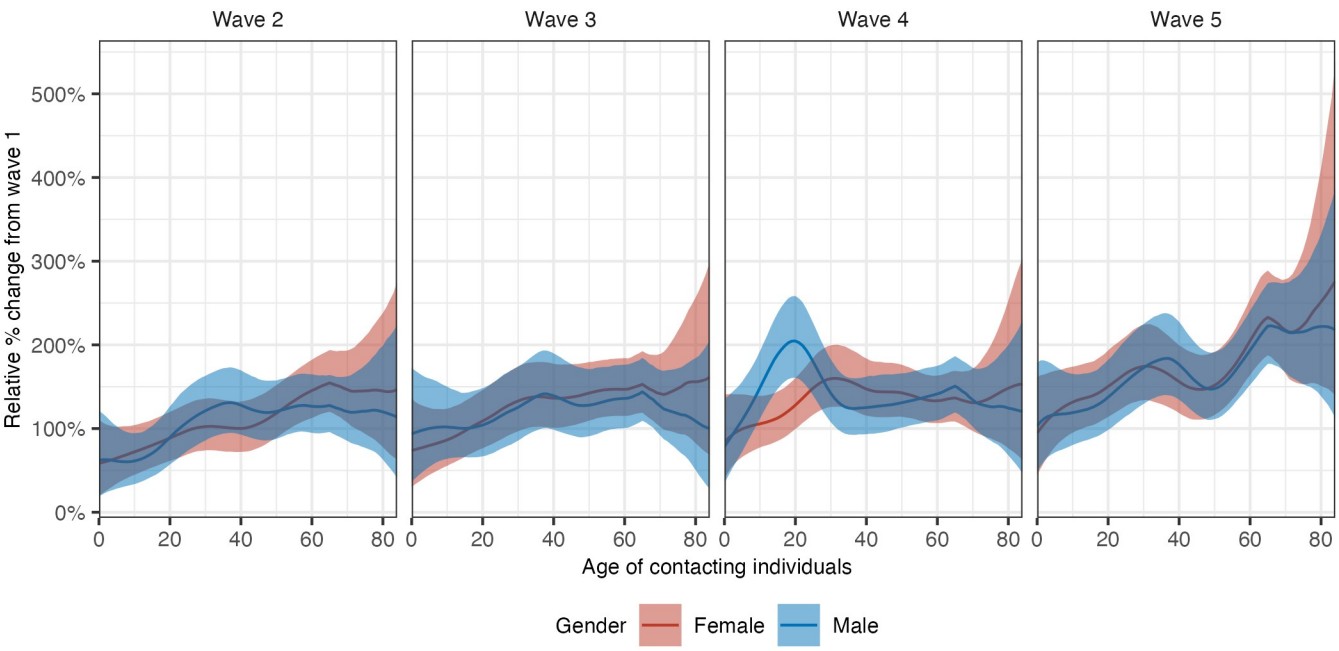

**Fig 8. Comparison of social contact intensities before and during the first months of the COVID-19 pandemic in Germany.** Solid lines represent posterior median estimates of the marginal contact intensities by age and gender of participants, and after adjusting for missing & aggregate contacts and reporting fatigue for the COVIMOD survey. Dashed lines represent posterior median estimates from the pre-pandemic POLYMOD study and, unlike the estimates reported in [1], also include missing & aggregate contacts. Shaded ribbons represent 95% credible intervals.

differences between the two genders, which suggests that the gender differences in contact patterns may arise from underlying gender-dependent social contact structures that are captured in the survey, rather than non-pharmaceutical interventions. Comparing waves 2 and 3 to wave 1, contact intensities tended to increase more in older age groups. For wave 4, we observe a sharp increase in contacts among men approximately 20 years of age. However, we find that this pattern is sensitive to data pre-processing criteria, as we explain below. Comparing wave 5 to wave 1, we found the strongest evidence that contact intensities did not increase homogeneously for all age groups over time and instead tended to increase more in older age groups.

## Differential rebound of age-specific social contacts

We next focus on characterising the dynamics in contact intensities for specific age groups. Intuitively, this corresponds to slicing the contact intensity matrices reported in Fig 6 across rows for a fixed column, and for brevity, we call these the "conditional" contact intensities. In Fig 10 (top), we illustrate the conditional contact intensities for individuals aged 10, 20, 35, and 70 years to represent the contact intensities of school children, young adults, the working, and the ageing population. We observe two peaks for participants aged 10 and 20: a larger sharp peak corresponding to contacts between peers and a shorter rounded peak with individuals approximately 45–50 years older, representing contacts with their parents. For participants aged 35, we observe an additional third peak with individuals aged 60 to 70, predominantly corresponding to contacts with their parents. Participants aged 70 generally mixed with individuals of a similar age, with some contact between individuals aged approximately 40 but almost no contact with individuals under 20. These core patterns of social contacts are present across all survey waves.

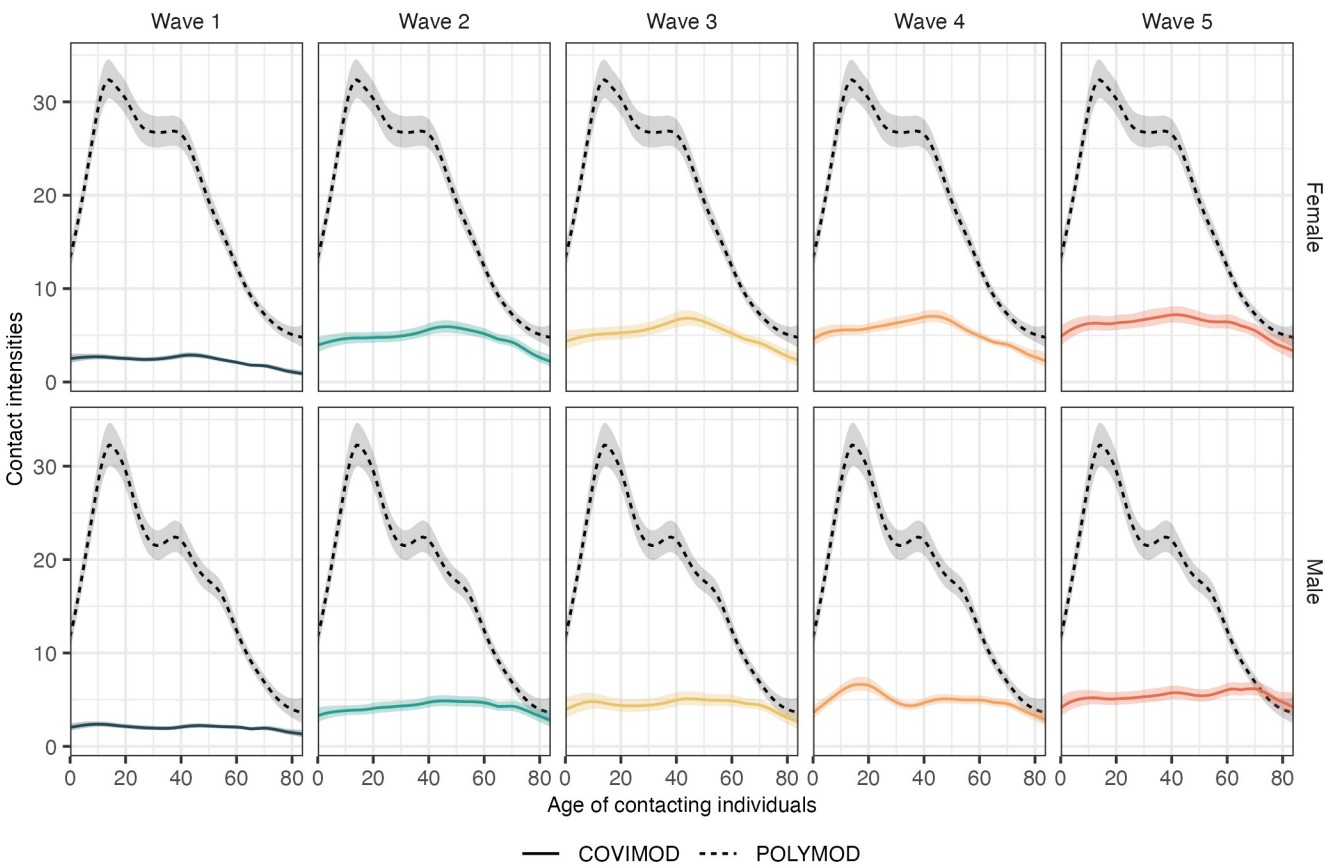

**Fig 9. Relative percentage change in marginal contact intensities relative to those in wave 1.** The red and blue lines represent posterior median estimates of the relative percentage change in marginal contact intensity estimates in women and men by the age of participants. Shaded ribbons represent 95% credible intervals.

Fig 10 (bottom) shows the ratio of the conditional contact intensities in survey waves 2, 3, 4 and 5 relative to those in wave 1. Ratios above 1 thus indicate increases in social contacts. We find that the increases in social contacts were not homogeneous by the age of contacted individuals. Focusing on wave 5 relative to wave 1, we find that for children aged 10, the conditional contact intensities in individuals of the same age and roughly aged 70 rose particularly strongly. For individuals aged 20, increases in their social contacts were more homogeneous, except for those with young children, which remained similar to those seen in wave 1. For individuals aged 35, increases in their social contacts were concentrated in slighter younger and all older individuals, but not their peers, reflecting that individuals in the 35-year age group retained social contact with their peers during intense non-pharmaceutical interventions in wave 1. Individuals aged 70 homogeneously increased their conditional contact intensities with younger individuals. These findings suggest that the age patterns in social contacts changed in a structured and complex manner after the first COVID-19 wave in Germany.

## Discussion

We developed the Bayesian rate consistency model in order to regain the ability to quantify and characterise social contact patterns at high age resolution from contemporary, longitudinal survey data on social contacts. The main contributions of our model-based approach to

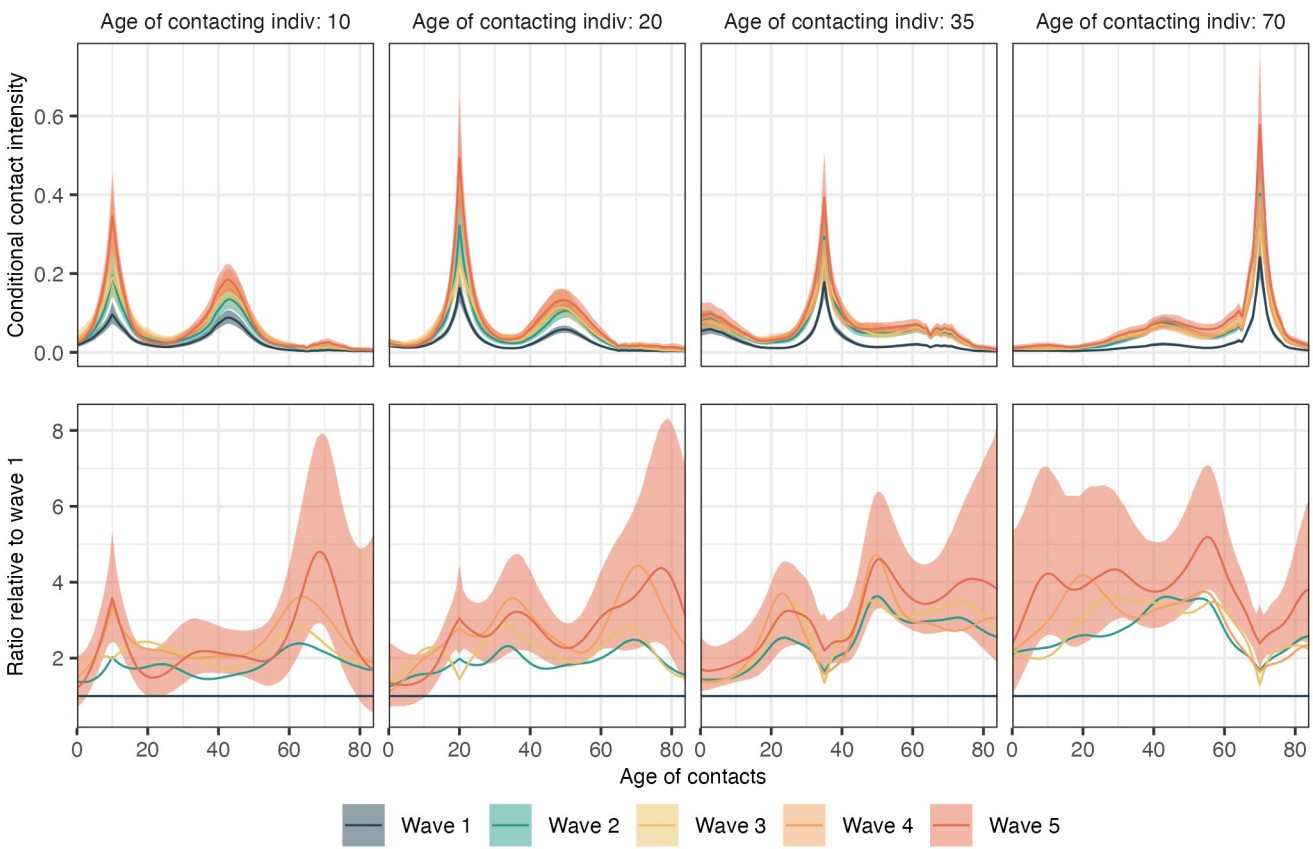

**Fig 10. Time evolution in age-specific social contact intensities.** The posterior median estimates for conditional contact intensities (top row) and relative change in the conditional contact intensities in waves 2 to 5 relative to those in wave 1 (bottom row) for individuals aged 10, 20, 35 and 70, respectively. Conditional contact intensities were aggregated across men and women. The colours represent different COVIMOD survey waves, and the shaded ribbons represent 95% credible intervals. We only show credible intervals for wave 5 to reduce overlaps and ease interpretation.

estimating the trends in human contact patterns are as follows. The Bayesian rate consistency model can reasonably accurately estimate high-resolution social contact patterns from data that aggregate the age of contacts into large age bands. The model also enables adjusting for the confounding effects due to the aggregated reporting of contacts and reporting fatigue in longitudinal social contact surveys. These advancements are particularly relevant to COVID-era social contact studies for which the age of contacts is reported in coarse age brackets and to which participants contributed to multiple survey waves [7, 13–15]. We draw from methodologies which serve as principal workhorses in spatial statistics to map the landscape of social contacts [33, 34] and incorporate a recently developed GP approximation technique to alleviate the computational bottleneck which often plagues such spatial models [16, 25].

In applying our model to the first five waves of the COVIMOD study, we gained insights into the age- and gender-specific social contact dynamics and their evolution over time at a resolution of 1-year age bands. We found relatively limited rebounds in contact intensities after lifting contact reduction measures and that social contact intensities remained substantially below pre-pandemic values until the end of our study period in July 2020 (Fig 7). These results are consistent with social contact survey data in England [13], Belgium [14], the Netherlands [15], and elsewhere. Relative to pre-pandemic estimates, the largest contact reductions occurred among children and young adults (Fig 7). Still, reductions were substantial across all age groups,

indicating the strong impact of non-pharmaceutical interventions on social interactions and successfully reduced infection risk during the first months of the COVID-19 pandemic [32].

Over the first five COVIMOD survey waves between May and July 2020, we observed structured changes in contact patterns, with social contact intensities rebounding more rapidly with the increasing age of individuals (Fig 7). These increases in social contacts were also far from uniform across population age bands of contacted individuals. Most increases tended to centre on the off-diagonals of the age-age social contact intensity matrix (Fig 6). These findings are hard to obtain without detailed inferences of social contact patterns by 1-year age bands [13–15]. We also noted that despite a continued increase in contact intensity, COVID-19 cases remained stable during the analysis period (Fig 1). This may be because contact counts remained much lower than pre-pandemic estimates despite the ease in non-pharmaceutical interventions [9]. Other protective measures such as face masks, hygiene regulations including surface disinfection, remote work, and warmer summer weather may also have contributed to keeping infections at bay [5]. These observations demonstrate that longitudinal data collection on social contacts and tailored tools to analyse these data are essential to characterise the impact of non-pharmaceutical interventions and the possible social contact routes through which pathogens can spread because the observed dynamics in social contacts do not follow easily predictable patterns.

Our work is not without limitations. First, some issues arise from the sampling methodology of the COVIMOD survey. Participants were recruited from an online panel for market research with email invitations, and although quota sampling was performed, the final samples were not fully representative of the population [9]. Previous work proposed using post-stratification weights to re-scale the data, but a sensitivity analysis did not reveal large differences between weighted and unweighted estimates [9]. Participants consisted only of those with internet access who possibly adhered more to social distancing rules as such a demographic is more likely to respond to health surveys. Participants received guidance only through the text within the questionnaires, which may have been misinterpreted, and participants may report more contacts in paper-based surveys than in an online survey [35]. Additionally, we truncated aggregate contact reports at 60, but different thresholds may lead to slight changes in the inference results (S1 Fig). Next, the difference-in-age parameterisation may not be appropriate if social contacts do not follow the pattern where high intensities lie on the contact matrix's main diagonal and sub-diagonals. This is relevant when investigators wish to conduct analyses for other contexts, e.g., work and transport, where contact patterns may not depend on age and age difference. However, it is easy for investigators to revert to the classical age-age parameterisation if they deem it more appropriate. We provide several template `Stan` model files in the accompanying GitHub repository. Furthermore, our fully Bayesian modelling framework is currently limited to analysing approximately 10 longitudinal survey rounds. While the recently proposed Hilbert Space Gaussian Process priors enable fast Bayesian inferences on cross-sectional data [16, 24, 25], additional research is needed to scale up the approach to survey data from 30 waves or more. Potential avenues include HSGP modelling over the time domain within our approach, Integrated Nested Laplace Approximations, penalised spline-based regressions or even variational autoencoders [2, 10, 36, 37]. Finally, and most importantly, the Bayesian rate consistency model requires participant age information to be reported by 1-year age bands. The exact age of participants is usually recorded without error but not necessarily made publicly available [7].

## Conclusion

In summary, we provide a new statistical method to strengthen the global pandemic preparedness toolkit, which enables epidemiologists and policymakers to obtain a clearer picture of how

infectious respiratory pathogens such as SARS-CoV-2 can spread through populations in near real-time. The Bayesian rate consistency model can analyse contemporary, longitudinal social contact survey data and estimate contact intensity by 1-year age bands. We validated the model on simulated social contact data for different scenarios and real-world data from Europe and Africa where contacts have been recorded in 1-year age bands, and any inference errors made by artificially coarsening the data can be assessed. The outputs—contact intensity matrices by gender and 1-year age band with uncertainty quantification—are central to estimating the effective reproduction number of pathogens in real-time with greater precision than currently possible from contact estimates by coarse age bands [6, 10, 13]. The outputs are also crucial for parameterising infectious disease models and more accurately forecasting cases, hospitalisations and deaths [32, 38, 39], and understanding the drivers of disease spread [38, 40]. Contact estimates from the first waves of the COVIMOD and CoMix social contact surveys [7] may be of particular interest as they represent the patterns during a time of stay-at-home orders across Europe and subsequent relaxation of non-pharmaceutical interventions, and could be used as templates of social contact patterns in future pandemic emergencies when such data are not immediately available. Vaccine allocation strategies also depend critically on understanding how pathogens spread through structured populations. Optimal allocations count the direct benefits of allocating life-saving vaccines to individuals with the highest fatality risk and the indirect benefits that accrue through prioritised allocations to population groups that would otherwise drive spread disproportionately through their contact and mixing patterns [41, 42]. We thus find the Bayesian rate consistency model promises to aid the understanding of contact behaviour, more realistic parameterisations of infectious disease models, and a deeper understanding of how infectious respiratory diseases are propagated through populations.

## Supporting information

**S1 Fig. The number of complete and aggregated contact reports by age, gender, and COVIMOD wave.** Pink bars represent missing & aggregate contacts, light blue bars represent non-household contacts, and dark blue bars represent household contacts. Missing & aggregate contacts were truncated at 60 (90th percentile in the primary data) to remove the effects of extreme outliers.
(JPEG)

**S2 Fig. Contribution of individually reported contacts with the known age of contact among all contacts reported.** Dark blue bars represent individually reported contacts for whom the age of the contact was specified by age bands. Pink bars represent reported contacts that were either reported in aggregate or individually without detail on the age or gender of the contact.
(JPEG)

**S3 Fig. The Average number of contacts for first-time and all participants by wave and contact type.** Red, light blue, green, and dark blue points represent the average number of household, non-household excluding aggregated & missing contacts, non-household including aggregated & missing contacts, and all contacts, respectively. Error bars represent 95% bootstrap confidence intervals.
(JPEG)

**S4 Fig. Simulation experiment results for the pre-COVID-19 scenario with different covariance kernels.** From top to bottom: results for the squared exponential kernel, results for the Matérn $\frac{5}{2}$ kernel, and results for the Matérn $\frac{3}{2}$ kernel. All experiments were run with HSGP using the difference-in-age parameterisation models with $M^1$ = 40 (Number of eigenfunctions

on the difference-in-age dimension) and $M^2 = 20$ (Number of eigenfunctions on the contacts' age dimension). The sample size was fixed at $N = 2000$.
(JPEG)

**S5 Fig. Simulation experiment results for the in-COVID19 scenario with different covariance kernels.** From top to bottom: results for the squared exponential kernel, results for the Matérn $\frac{5}{2}$ kernel, and results for the Matérn $\frac{3}{2}$ kernel. All experiments were run with HSGP using the difference-in-age parameterisation models with $M^1 = 40$ (Number of eigenfunctions on the difference-in-age dimension) and $M^2 = 20$ (Number of eigenfunctions on the contacts' age dimension). The sample size was fixed at $N = 2000$.
(JPEG)

**S6 Fig. Simulated data, estimated contact intensity matrix, and estimated marginal contact intensities of different sample sizes for the pre-COVID-19 scenario.** (Top row) Simulated contact data for the pre-COVID-19 scenario. (Middle row) Posterior median contact intensity estimates. (Bottom row) Posterior median marginal contact intensity estimates. Results were obtained using the Bayesian rate consistency model with difference-in-age parameterisation with $M^1 = 30$ and $M^2 = 20$.
(PNG)

**S7 Fig. Simulated data, estimated contact intensity matrix, and estimated marginal contact intensities of different sample sizes for the in-COVID-19 scenario.** (Top row) Simulated contact data for the in-COVID-19 scenario. (Middle row) Posterior median contact intensity estimates. (Bottom row) Posterior median marginal contact intensity estimates. Results were obtained using the Bayesian rate consistency model with difference-in-age parameterisation with $M^1 = 30$ and $M^2 = 20$.
(PNG)

**S8 Fig. Trace plots of the parameters with the smallest effective sample size and maximum $\hat{R}$ convergence diagnostic statistic.** (Top row) Trace plot for parameter with the lowest effective sample size (1892), `gp_rho_2[5, 2]`: the length-scale parameter of the Gaussian process over the contact age dimension of the Female-Female contact matrix for wave 5. (Bottom row) Trace plot for the parameter with the largest $\hat{R}$ convergence diagnostic statistic (1.01), `z [1, 21, 6]`: the coefficient for the 21$^{st}$ HSGP basis function over the difference-in-age dimension of the Female-Female contact matrix for wave 1.
(PNG)

**S9 Fig. Empirical and estimated social contact intensity patterns for COVIMOD wave 2.** (Top row) Crude empirical social contact intensity patterns, with crude contact intensities above a value of 3 truncated for visualisation purposes. There are some age groups with no participants, and they are represented by white vertical columns. (Middle row) Contact intensity patterns as estimated by the `socialmixr` R package [11]. (Bottom row) Contact intensity patterns are given by our Bayesian model.
(JPEG)

**S10 Fig. Empirical and estimated social contact intensity patterns for COVIMOD wave 3.** (Top row) Crude empirical social contact intensity patterns, with crude contact intensities above a value of 3 truncated for visualisation purposes. There are some age groups with no participants, and they are represented by white vertical columns. (Middle row) Contact intensity patterns as estimated by the `socialmixr` R package [11]. (Bottom row) Contact intensity

patterns are given by our Bayesian model.
(JPEG)

**S11 Fig. Empirical and estimated social contact intensity patterns for COVIMOD wave 4.**
(Top row) Crude empirical social contact intensity patterns, with crude contact intensities
above a value of 3 truncated for visualisation purposes. There are some age groups with no par-
ticipants, and they are represented by white vertical columns. (Middle row) Contact intensity
patterns as estimated by the `socialmixr` R package [11]. (Bottom row) Contact intensity
patterns are given by our Bayesian model.
(JPEG)

**S12 Fig. Empirical and estimated social contact intensity patterns for COVIMOD wave 5.**
(Top row) Crude empirical social contact intensity patterns, with crude contact intensities
above a value of 3 truncated for visualisation purposes. There are some age groups with no par-
ticipants, and they are represented by white vertical columns. (Middle row) Contact intensity
patterns as estimated by the `socialmixr` R package [11]. (Bottom row) Contact intensity
patterns are given by our Bayesian model.
(JPEG)

**S13 Fig. Comparison of marginal contact intensity estimates: Final longitudinal model
adjusting for reporting fatigue v.s. a cross-sectional model with repeating participants
excluded from the data.** The red lines and ribbons show the posterior median estimates and
95% credible interval from the final longitudinal model on the entire COVIMOD dataset,
adjusting for reporting fatigue. The green lines and ribbons display the estimated median and
95% credible interval from a cross-sectional model fitted on COVIMOD wave 4 data, exclud-
ing participants who answered the survey in previous waves.
(JPEG)

**S14 Fig. Ratio of female to male marginal contact intensities for POLYMOD and COVI-
MOD.** Lines represent posterior median estimates of the female-to-male marginal contact
intensity ratios, i.e., $m_a^F/m_a^M$. A ratio of 1 (dashed lines) indicates no difference in contact
intensities between genders. Shaded ribbons represent 95% credible intervals.
(JPEG)

**S1 Text. The construction of simulated social contact patterns.** A detailed description of
how the contact intensity patterns used in the simulation experiments are generated.
(PDF)

**S1 Table. Comparison of different covariance kernels and number of basis functions for
HSGP models.** Results were obtained with models using the difference-in-age parameterisa-
tion. The sample size was fixed at $N = 2000$ throughout. $M^1$: The number of HSGP basis func-
tions on the difference-in-age dimension. $M^2$: The number of HSGP basis functions on the
contacts' age dimension. [a]Mean absolute error, [b]Expected log posterior density, [c]Posterior pre-
dictive check, [d]Median runtime, [e]pre-COVID19, [f]in-COVID19 scenario.
(PDF)

## Acknowledgments

The authors acknowledge Christopher Jarvis, Kevin Van Zandvoort, Amy Gimma, John
Edmunds, and the entire CoMix team for allowing the COVIMOD team to use an adapted ver-
sion of the CoMix questionnaire for COVIMOD and for their cooperation. The authors also
thank the team at IPSOS-Mori for their work on implementing the COVIMOD survey.

## Author Contributions

**Conceptualization:** Shozen Dan, Veronika K. Jaeger, Oliver Ratmann.

**Data curation:** Shozen Dan, Veronika K. Jaeger, André Karch.

**Formal analysis:** Shozen Dan.

**Funding acquisition:** Veronika K. Jaeger, André Karch, Oliver Ratmann.

**Investigation:** Shozen Dan, Yu Chen, Oliver Ratmann.

**Methodology:** Shozen Dan, Yu Chen, Yining Chen, Melodie Monod, Samir Bhatt, Oliver Ratmann.

**Project administration:** Shozen Dan.

**Resources:** Veronika K. Jaeger.

**Software:** Shozen Dan, Yu Chen.

**Supervision:** Shozen Dan, Oliver Ratmann.

**Validation:** Shozen Dan.

**Visualization:** Shozen Dan, Yu Chen, Oliver Ratmann.

**Writing – original draft:** Shozen Dan, Yu Chen, Oliver Ratmann.

**Writing – review & editing:** Shozen Dan, Yu Chen, Veronika K. Jaeger, Samir Bhatt, André Karch, Oliver Ratmann.

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
