## [Decision Letter · Decision Letter 0]

4 Feb 2023

Dear Mr. Dan,

Thank you very much for submitting your manuscript "Estimating fine age structure and time trends in human contact patterns from coarse contact data: the Bayesian rate consistency model" for consideration at PLOS Computational Biology.

As with all papers reviewed by the journal, your manuscript was reviewed by members of the editorial board and by several independent reviewers. In light of the reviews (below this email), we would like to invite the resubmission of a significantly-revised version that takes into account the reviewers' comments.

We cannot make any decision about publication until we have seen the revised manuscript and your response to the reviewers' comments. Your revised manuscript is also likely to be sent to reviewers for further evaluation.

Sincerely,

Virginia E. Pitzer, Sc.D.

Section Editor

PLOS Computational Biology

Reviewer's Responses to Questions

**Comments to the Authors:**

Reviewer #1: This manuscript is attractive and well-written but methodologically under-developed. The full-Bayesian model should be expanded to account for temporal trends. Currently, f_t, tau_t, alpha_t, l_t are treated as a priori independent, but use of splines or GPs or linear models would help borrow information across survey windows and explicitly account for temporal trends. The same goes for the time varying reporting effects.

Major Comments:

1) Regarding second sentence of section "Data Processing". The method of imputing children's ages seems more like adding noise than actual imputation (which usually uses ancillary data to predict missing data). Since you're already working within a Bayesian framework, wouldn't it make more sense to infer these ages as latent variables? Otherwise, it is hard to understand the benefit of adding noise.

2) The choice of directly including (8) in (9) seems arbitrary. Wouldn't it be reasonable to include any function that favors Y and penalizes T?

3) I find the HSGP terminology a little bit troubling. Isn't it more accurate to call this a truncated Karhunen-Loeve expansion? Is HSGP a rebranding or is it really different?

Minor Comments:

1) I don't think the stringency index is ever defined or its relevance is ever stated. It is cited, but I shouldn't have to look it up elsewhere.

2) Sentence beginning "The overdispersion parameter" is not a sentence.

3) "i. e."

4) What is the main point of the second paragraph within the "Difference in Age Parameterization" section? Is it "this works better and is worth the additional computational cost"? Please clarify.

Reviewer #2: This paper introduces a novel method to analyse contact survey data where the age characteristic of the contact has only been reported as an age group. The method allows estimation of contact matrices at a finer age scale. The resulting matrices remind of approaches where a contact matrix of aggregate age groups has been smoothed (e.g. https://doi.org/10.1186/1471-2334-9-187), but the method presented here arguably make better use of the precise recording of contactor ages. In addition, the authors aim to adjust their estimates by reporting fatigue, and for data collected at an aggregate level. This is then applied to analyse social contact data from Germany.

The authors used zero-mean two-dimensional Gaussian Processes to estimate the contact intensities between two ages. I am not familiar enough with this method to accurately review this technique. I have assumed that it is valid and appropriate in my review, but would recommend a specialist review to look at this aspect of the method.

Overall, this is a very interesting paper and a useful addition to the literature on contact matrices. I do have a number of comments, mainly on i) the adjustment for the aggregated number of contacts, and ii) the simulations used to validate the model.

Major comments:

i) Adjustment of the aggregated number of contacts

Both the incidence of cases and stringency index remained relatively stable between waves 2 to 5 (figure 1). However, your model estimates a gradual increase in contacts between these periods (figures 5, 6, and 8). This is contradictory, as you would roughly expect changes in the force of infection and incidence to be proportional to the change in the number of cases. In Figure S1, I would expect non-household contacts to increase as NPIs are relaxed. However, they remain stable, while mainly the aggregated contacts seem to increase over time. Part of that is artificial, because they could only be reported from wave 3 onwards (line 89 – please explain why there are also aggregate contacts in wave 2 in your figure S1). But I would expect to see a similar increase in non-household contacts if there was indeed a true change in behaviour increase. I have some doubts about the data quality and accuracy of the reported aggregate number of contacts. Direct contacts are relatively clearly defined (physical touch and/or a short in-person conversation), but people can also have other in-person contacts that do not necessarily fit this definition (e.g. shop clerks interacting with customers). These other contacts are not as clearly defined, but often referred to as indirect contacts. Many surveys, including POLYMOD, allowed participants to roughly estimate such indirect contacts. They are probably less important for transmission than direct contacts (no/fewer words spoken, usually of shorter duration or larger physical distance). The aggregate contacts referred to in this paper may also be an example of such indirect contacts (I could not see the actual wording of the questions asked), and I think that this should be explored further:

- At a minimum, could you report the proportion of aggregate to individually reported contacts in each age and wave?

- In lines 93, 140, 205: It is not clear how participants with aggregate contacts are treated. Are the contacts added in Y_ab? Or do only the participants add to N_a, but their aggregate contacts not to Y_ab? If a person aged a reports 20 aggregate contacts, does that mean 20 is added to T_ta? Please clarify this in the text.

If they are added to T_ta (and thus given equal weight as contacts in Y_ab), it would be useful to explore treating them differently, either by excluding them from the analysis, or by giving them a lower weight than the direct contacts (it can potentially be fitted as a parameter in your model, you will have some information if you assume people of similar age have on average the same number of daily contacts, and there are sufficient people with/without aggregate contacts).

ii) Validation of the model through simulated contact data

The simulated datasets seem to generate realistic contact matrices with strong age-assortative mixing and off-diagonals representing child-parent contacts. The model seems to be able to estimate this matrix quite well, where the difference-in-age method is better than the age-age method. However, that is perhaps not surprising, as the data is also generated to make sure that there are strong difference-in-age patterns.

Real-world data is a lot messier, and it would be useful to see how well the model performs against a real-world dataset where the age of the contact is known. POLYMOD has the age in years (as an integer) for both participants and most contacts. It would be useful to understand how well the model can estimate the age-granular POLYMOD matrix (assuming that it is true) from POLYMOD data converted to data in age groups. In addition, mixing patterns can look very different in populations that do not follow a rectangular age distribution (such as Germany or the UK), but where the age distribution is pyramid-shaped as is the case in many countries with high mortality rates (e.g. Kenya or Malawi). In these populations, age-assortative mixing is often less apparent in matrices as contacts between children dominate the overall contacts. It would be very useful to understand how well the model behaves for matrices in these populations.

Minor comments:

17: More difficult compared to what exactly?

22, 40: The authors write that participants record contact ages in large age categories COVID-era studies to facilitate reporting. In many surveys, this was a restriction decided on by the survey implementor (the online market research platform) for ethical and privacy reasons. Note that it is not unique to COVID-era studies, e.g. POLYMOD also allowed participants to provide an age range if the exact age of a contact was not known (which is common).

120, 146: Why did you cut your matrices off above age 84? Are participants and contacts above these ages grouped together, and should the final age group be interpreted as a single 84+ group?

133: when imputing the age of the child, did you create multiple datasets to fit to? Did you redraw the age in each MCMC iteration?

138: Unclear what is meant by a missing entry. I assume it means that not a single participant of age a and gender g reports any contact with contacts of age b and gender h, but this is not clear.

156, 161: be consistent when describing contact rates, e.g. make it clear that this is the rate per unit time (probably day)

Useful to double check notations. Especially M, F and g, h subscripts are used interchangeable, which is not always necessary.

187, 191: It may be better to speak of survey wave rather than time or survey time

283: I can’t find any trace plots in the supplement. Not possible to assess mixing and convergence of chains. Would be useful to have a table with all summary statistics of the chains as well (for the final model).

293: small typo: studies is duplicate

Table 1: it is hard to understand how the accuracy of the method changes from the summary statistics alone. Could these matrices also be plotted in the supplement?

368: did you use the same population distribution in adjusting the socialmixr estimates for symmetry as has been used in your model (your P_a parameter)? This will (slightly) affect the final estimated contact matrix.

370: contacts with missing age information are imputed by sampling. Do the socialmixr figures show a single (random) iteration of this sampling? Similarly, how are you selecting the final fitted parameters to generate your shown contact matrices from your posterior distribution?

Figure 4: is it possible to show the intensity for the top and middle row at the same scale as the bottom row? E.g. by dividing the value for each cell for each integer age in the contact age group, and copying this value for each age in the participant age group? This also makes the comparison between age groups a bit fairer, as some are in 5y age bands and other are in 10y age bands.

Figure 5: how are the central estimates calculated? Median of posterior samples?

Figure 5, 192: I agree with the attempted adjustment for reporting fatigue. However, it would be useful to understand how well your method is able to make this adjustment. Your rho parameter essentially rescales all contacts for people who have completed >1 survey. Could you compare the adjusted estimates in figure 5 to a model fit where all data is restricted to new participants?

Figure 7: The POLYMOD estimates seem very high. The mean contacts in the Mossong et al paper are around 10-15 for most age groups, and similar for Germany. Could you double check these? Have you included the estimated indirect contacts in the POLYMOD estimates? Because of different survey methodologies (and changes in behaviour), the differences between POLYMOD and COVIMOD are probably not only due to the pandemic, which should be accounted for when comparing the two surveys.

484: This is figure 7.

485: I don’t think you can conclude that there has been a sustained behavioural change compared to pre-pandemic values, as the two surveys are not directly comparable.

528: I assume this is a computational limitation? Arguably, the contact patterns by age are much more important than those by/between gender. Combining the two genders may allow fitting to a larger number of survey waves.

**Have the authors made all data and (if applicable) computational code underlying the findings in their manuscript fully available?**

Reviewer #1: Yes

Reviewer #2: Yes

PLOS authors have the option to publish the peer review history of their article (what does this mean?). If published, this will include your full peer review and any attached files.

Reviewer #1: No

Reviewer #2: No
---

## [Decision Letter · Decision Letter 1]

17 May 2023

Dear Mr. Dan,

We are pleased to inform you that your manuscript 'Estimating fine age structure and time trends in human contact patterns from coarse contact data: the Bayesian rate consistency model' has been provisionally accepted for publication in PLOS Computational Biology.

Reviewer 2 also noted a few minor typos that you may wish to address at this stage.

Best regards,

Virginia E. Pitzer, Sc.D.

Section Editor

PLOS Computational Biology

Lucy Houghton

Staff

PLOS Computational Biology

Reviewer's Responses to Questions

**Comments to the Authors:**

Reviewer #1: I'm satisfied.

Reviewer #2: I would like to thank the authors for their work and additional time in running the additional analyses. I am very happy with their results, and with the edits made. All my comments from my first review have been answered, and I have no further comments.

I did notice a few typos that I wanted to highlight to the authors:

Line 533: 20202

Line 614: S2 Fig is duplicated

Line 619: A is capitalized

**Have the authors made all data and (if applicable) computational code underlying the findings in their manuscript fully available?**

Reviewer #1: Yes

Reviewer #2: Yes

PLOS authors have the option to publish the peer review history of their article (what does this mean?). If published, this will include your full peer review and any attached files.

Reviewer #1: No

Reviewer #2: **Yes: **Kevin van Zandvoort

---

## [Editor Report · Acceptance letter]

1 Jun 2023

PCOMPBIOL-D-22-01544R1 

Estimating fine age structure and time trends in human contact patterns from coarse contact data: the Bayesian rate consistency model

Dear Dr Dan,

I am pleased to inform you that your manuscript has been formally accepted for publication in PLOS Computational Biology. Your manuscript is now with our production department and you will be notified of the publication date in due course.

With kind regards,

Zsofia Freund
